# WILDFUSION: LEARNING 3D-AWARE LATENT DIFFUSION MODELS IN VIEW SPACE

**Katja Schwarz**[1,†]**, Seung Wook Kim**[2,3,4]**, Jun Gao**[2,3,4]**, Sanja Fidler**[2,3,4]**, Andreas Geiger**[1]**, Karsten Kreis**[2]
[1]University of Tübingen, [2]NVIDIA, [3]Vector Institute, [4]University of Toronto

## ABSTRACT

Modern learning-based approaches to 3D-aware image synthesis achieve high photorealism and 3D-consistent viewpoint changes for the generated images. Existing approaches represent instances in a shared canonical space. However, for in-the-wild datasets a shared canonical system can be difficult to define or might not even exist. In this work, we instead model instances in *view space*, alleviating the need for posed images and learned camera distributions. We find that in this setting, existing GAN-based methods are prone to generating flat geometry and struggle with distribution coverage. We hence propose *WildFusion*, a new approach to 3D-aware image synthesis based on latent diffusion models (LDMs). We first train an autoencoder that infers a compressed latent representation, which additionally captures the images' underlying 3D structure and enables not only reconstruction but also novel view synthesis. To learn a faithful 3D representation, we leverage cues from monocular depth prediction. Then, we train a diffusion model in the 3D-aware latent space, thereby enabling synthesis of high-quality 3D-consistent image samples, outperforming recent state-of-the-art GAN-based methods. Importantly, our 3D-aware LDM is trained without any direct supervision from multiview images or 3D geometry and does not require posed images or learned pose or camera distributions. It directly learns a 3D representation without relying on canonical camera coordinates. This opens up promising research avenues for scalable 3D-aware image synthesis and 3D content creation from in-the-wild image data. See https://katjaschwarz.github.io/wildfusion/ for videos of our 3D results.

## 1 INTRODUCTION

High-quality 2D and 3D content creation is of great importance for virtual reality, animated movies, gaming and robotics simulation. In the past years, deep generative models have demonstrated immense potential, enabling photorealistic image synthesis at high resolution (Goodfellow et al., 2014; Karras et al., 2020; 2021; Rombach et al., 2021; Dhariwal & Nichol, 2021b; Sauer et al., 2022). Recently, 3D-aware generative models advanced image synthesis to view-consistent, 3D-aware image generation (Schwarz et al., 2020; Chan et al., 2021; Gu et al., 2022; Jo et al., 2021; Xu et al., 2022; Zhou et al., 2021b; Zhang et al., 2021; Or-El et al., 2022; Xu et al., 2021; Pan et al., 2021; Deng et al., 2022b; Xiang et al., 2023a; Schwarz et al., 2022; Chan et al., 2022). They generate images with explicit control over the camera viewpoint. Importantly, these approaches do not require 3D training data or multiview supervision, which is costly or impossible to obtain for large-scale real-world data.

While existing 3D-aware generative models achieve high photorealism and 3D-consistent viewpoint control, the vast majority of approaches only consider single-class and aligned data like human faces (Karras et al., 2020) or cat faces (Choi et al., 2020). The reason for this is that existing methods assume a shared canonical coordinate system to represent 3D objects. As a consequence, they require either poses from an off-the-shelf pose estimator (Chan et al., 2022; Schwarz et al., 2022; Gu et al., 2022; Xiang et al., 2023a) or assume, and sometimes learn to refine, a given pose distribution (Schwarz et al., 2020; Niemeyer & Geiger, 2021b; Shi et al., 2023; Skorokhodov et al., 2023; 2022). In contrast, in-the-wild images typically have no clearly defined canonical camera system and camera poses or pose distributions are not available or very challenging to obtain.

We propose to instead model instances in *view space*: Our coordinate system is viewer-centric, i.e., it parameterizes the space as seen from the camera's point of view. This removes the need for camera

---

†Part of the work was done during an internship at NVIDIA.

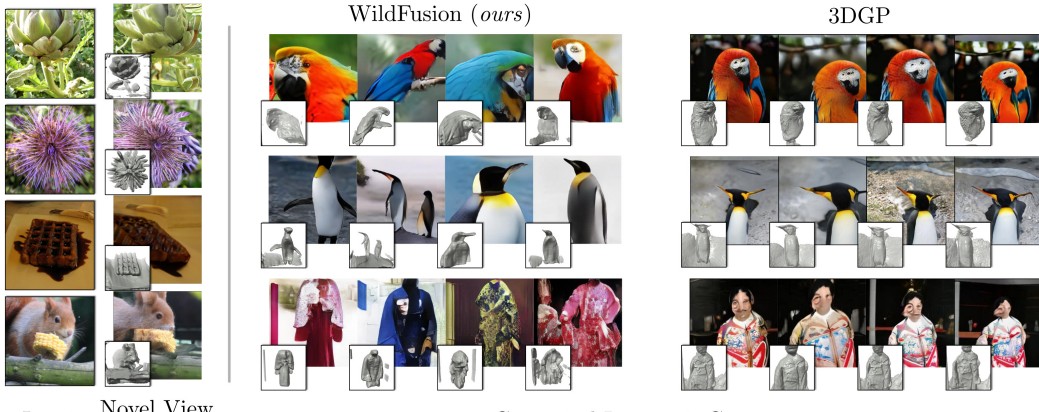

Figure 1: **WildFusion:** *Left:* Input images, novel views and geometry from first-stage autoencoder. *Right:* Novel samples and geometry from our second-stage latent diffusion model and 3DGP (Skorokhodov et al., 2023) for the ImageNet classes "macaw" (*top*), "king penguin" (*middle*), "kimono" (*bottom*). Included videos for more results.

poses and a priori camera pose distributions, unlocking 3D-aware image synthesis on unaligned, diverse datasets. We identify crucial challenges for training in view space: For complex datasets, without a shared canonical representation, existing techniques are prone to generating poor 3D representations and the GAN-based methods struggle with distribution coverage (see Fig. 2, Table 2).

To prevent generating flat 3D representations, we leverage cues from monocular depth prediction. While monocular depth estimators are typically trained with multi-view data, we leverage an off-the-shelf pretrained model, such that our approach does not require any *direct* multi-view supervision for training. Recent works (Bhat et al., 2023; Eftekhar et al., 2021; Ranftl et al., 2020; Miangoleh et al., 2021) demonstrate high prediction quality and generalization ability to in-the-wild data and have been successfully applied to improve 3D-reconstruction (Yu et al., 2022). To ensure distribution coverage on more diverse datasets, we build our approach upon denoising diffusion-based generative models (DDMs) (Ho et al., 2020; Sohl-Dickstein et al., 2015; Song et al., 2020). DDMs have shown state-of-the-art 2D image synthesis quality and offer a scalable and robust training objective (Nichol & Dhariwal, 2021; Nichol et al., 2022b; Rombach et al., 2021; Dhariwal & Nichol, 2021a; Ramesh et al., 2022; Saharia et al., 2022; Balaji et al., 2022; Ho et al., 2022). More specifically, we develop a 3D-aware generative model based on latent diffusion models (LDMs) (Rombach et al., 2021; Vahdat et al., 2021). By training an autoencoder first and then modeling encoded data in a compressed latent space, LDMs achieve an excellent trade-off between computational efficiency and quality. Further, their structured latent space can be learnt to capture a 3D representation of the modeled inputs, as we show in this work.

Our 3D-aware LDM, called *WildFusion*, follows LDMs' two-stage approach: First, we train a powerful 3D-aware autoencoder from large collections of unposed images without multiview supervision that simultaneously performs both compression *and* enables novel-view synthesis. The autoencoder is trained with pixel-space reconstruction losses on the input views and uses adversarial training to supervise novel views. Note that by using adversarial supervision for the novel views, our autoencoder is trained for novel-view synthesis without the need for multiview supervision, in contrast to previous work (Watson et al., 2022; Chan et al., 2023; Liu et al., 2023). Adding monocular depth cues helps the model learn a faithful 3D representation and further improves novel-view synthesis. In the second stage, we train a diffusion model in the compressed and 3D-aware latent space, which enables us to synthesize novel samples and turns the novel-view synthesis system, i.e., our autoencoder, into a 3D-aware generative model. We validate WildFusion on multiple image generation benchmarks, including ImageNet, and find that it outperforms recent state-of-the-art 3D-aware GANs. Moreover, we show that our autoencoder is able to directly synthesize high-quality novel views for a given single image and performs superior compared to recent GAN-based methods, which usually require an inversion process to embed a given image into their latent space (Abdal et al., 2019; Richardson et al., 2021; Tov et al., 2021; Zhu et al., 2020; Roich et al., 2023). Further, in contrast to inversion methods, our autoencoder is trained in a single stage and does not require a pretrained 3D-aware GAN as well as elaborate and often slow techniques for latent optimization.

Main contributions: **(i)** We remove the need for posed images and a priori camera pose distributions for 3D-aware image synthesis by modeling instances in *view space* instead of canonical space. **(ii)**

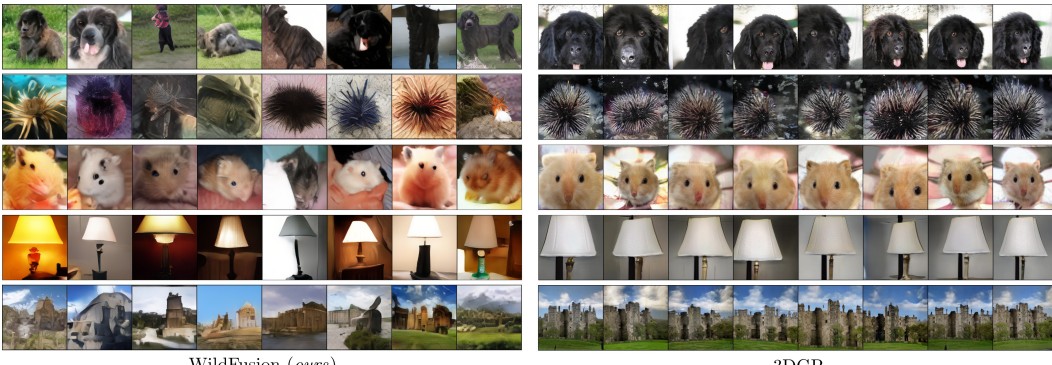

WildFusion (*ours*)                                                3DGP

Figure 2: **Sample Diversity:** Generated samples on ImageNet. Rows indicate class; columns show uncurated random samples. While WildFusion generates diverse samples due to its diffusion model-based framework (*left*), the GAN-based 3DGP (Skorokhodov et al., 2023) has very low intra-class diversity (mode collapse, *right*).

We learn a powerful 3D-aware autoencoder from unposed images without multiview supervision that simultaneously performs compression, while inferring a 3D representation suitable for novel-view synthesis. **(iii)** We show that our novel 3D-aware LDM, WildFusion, enables high-quality 3D-aware image synthesis with reasonable geometry and strong distribution coverage, achieving state-of-the-art performance in the unposed image training setting, which corresponds to training on in-the-wild image data. Moreover, we can more efficiently perform novel view synthesis for given images than common GAN-based methods and explore promising 3d-aware image manipulation techniques. We hope that WildFusion paves the way towards scalable and robust in-the-wild 3D-aware image synthesis.

## 2  RELATED WORK

We briefly provide the most relevant related works in this section, and present a comprehensive discussion of the theoretical fundamentals and the related literature, including concurrent works, in App. B.

Most works on 3D-aware image synthesis rely on GANs (Goodfellow et al., 2014) and focus on aligned datasets with well-defined pose distributions. For instance, POF3D (Shi et al., 2023) infers camera poses and works in a canonical view space; it has been used only for datasets with simple pose distributions, such as cat and human faces. To enable training on more complex datasets, 3DGP (Skorokhodov et al., 2023) proposes an elaborate camera model and learns to refine an initial prior on the pose distribution. Specifically, 3DGP predicts the camera location in a canonical coordinate system per class and sample-specific camera rotation and intrinsics. This assumes that samples within a class share a canonical system, and we observe that learning this complex distribution can aggravate training instability. Further, the approach needs to be trained on heavily filtered training data. In contrast, WildFusion can generate high-quality and diverse samples even when trained on the entire ImageNet dataset without any filtering (see Sec. 4.2). Moreover, we use EG3D's triplanes and their dual discriminator to improve view consistency (Chan et al., 2022). Note that in contrast to POF3D, 3DGP, EG3D, and the vast majority of 3D-aware image generative models, WildFusion is not a GAN. GANs are notoriously hard to train (Mescheder et al., 2018) and often do not cover the data distribution well (see mode collapse in 3DGP, Fig. 2). Instead, we explore 3D-aware image synthesis with latent diffusion models for the first time.

Concurrently with us, IVID (Xiang et al., 2023b) trains a 2D diffusion model that first synthesizes an initial image and subsequently generates novel views conditioned on it. However, its iterative generation is extremely slow because it requires running the full reverse diffusion process for every novel view. Instead, WildFusion only runs the reverse diffusion process once to generate a (latent) 3D representation. Another concurrent work, VQ3D (Sargent et al., 2023), also proposes an autoencoder architecture, but uses sequence-like latent variables and trains an autoregressive transformer in the latent space whereas we train a diffusion model on latent feature maps.

## 3  WILDFUSION

Our goal is to design a 3D-aware image synthesis framework that can be trained using unposed in-the-wild images. We base our framework, WildFusion, on LDMs (Rombach et al., 2021; Vahdat et al., 2021) for several reasons: **(i)** Compared to diffusion in the original data space, they offer excellent computational efficiency due to their compressed latent space. **(ii)** Diffusion models use a

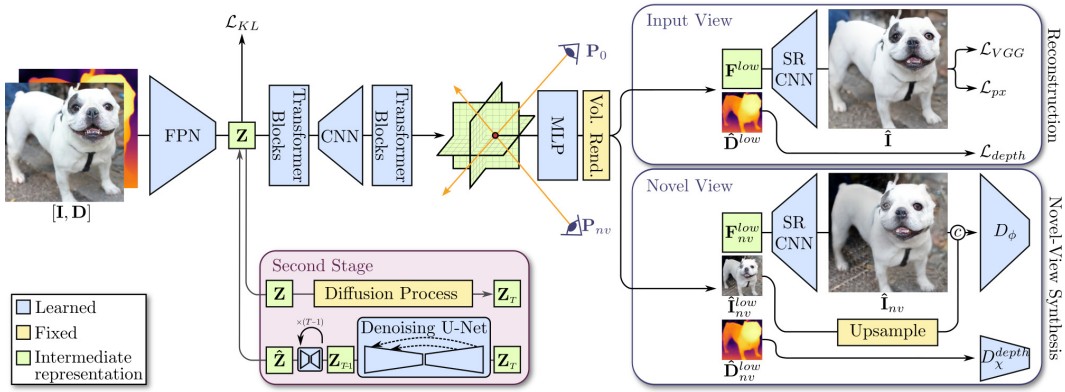

Figure 3: **WildFusion Overview:** In the first stage, we train an autoencoder for both compression and novel-view synthesis. A Feature Pyramid Network (FPN) (Lin et al., 2017) encodes a given unposed image $\mathbf{I}$ into an 3D-aware latent representation $\mathbf{Z}$, constructed as a 2D feature grid. A combination of transformer blocks and a CNN then decode $\mathbf{Z}$ into a triplane representation, which is rendered from both the input view $\mathbf{P}_0$ and a novel view $\mathbf{P}_{nv}$. As we model instances in view space, $\mathbf{P}_0$ is a fixed, pre-defined camera pose. The input view is supervised with reconstruction losses. Adversarial training provides supervision for novel views. In the second stage, a latent diffusion model is trained on the learned latent space to obtain a 3D-aware generative model.

robust objective that offers sample diversity and does not suffer from problems that plague GANs such as mode collapse. **(iii)** Most importantly, one can construct a latent space that can be trained to not only perform compression but also to learn a powerful 3D representation for novel view synthesis, as we demonstrate in this work. Fig. 3 shows an overview over WildFusion. It consists of two training stages (Rombach et al., 2021; Esser et al., 2021). In the first stage, our new autoencoder learns a compressed and abstracted latent space suitable for reconstruction and novel view synthesis from single view training data. In the second stage, a latent diffusion model is trained on the latent representation from the first stage autoencoder to obtain a full generative model.

## 3.1 AUTOENCODER FOR COMPRESSION AND NOVEL-VIEW SYNTHESIS

Following the LDM framework (Rombach et al., 2021), we first train an autoencoder that encodes training data into latent representations. Unlike LDMs, however, where the task of the autoencoder is simply to compress and reconstruct the inputs, our setting is more complex, as the autoencoding model must also learn a 3D representation of the data such that it can infer reasonable novel views from a single input image. The capability for novel-view synthesis will be used later by the diffusion model to perform 3D-aware image synthesis with 3D-consistent viewpoint changes. However, as no multiview or explicit 3D geometry supervision is available, this novel-view synthesis task is highly under-constrained and non-trivial to solve. To aid this process, we provide additional cues about the geometry in the form of monocular depth supervision from a pre-trained network (Bhat et al., 2023).

Specifically, we concatenate a given image $\mathbf{I} \in \mathbb{R}^{3 \times H \times W}$ with its estimated monocular depth $\mathbf{D} \in \mathbb{R}^{1 \times H \times W}$ channel-wise and encode them into a compressed latent representation $\mathbf{Z} \in \mathbb{R}^{c \times h \times w}$ via an encoder. As the encoder must infer a latent representation that encodes the underlying 3D object or scene of the input image, we found it beneficial to provide both $\mathbf{I}$ and $\mathbf{D}$ as input (see Table 4). We choose a Feature Pyramid Network (FPN) (Lin et al., 2017) architecture due to its large receptive field. For LDMs, latent space compression is crucial to train a diffusion model efficiently in the second stage. At input resolution $256 \times 256$ pixels, we use $c=4$, $h=w=32$ as in Rombach et al. (2021).

The decoder predicts a feature field from the compressed latent code $\mathbf{Z}$, which can be rendered from arbitrary viewing directions. The feature field is represented with triplanes. In contrast to previous works (Chan et al., 2022; Skorokhodov et al., 2022), our triplane representation is constructed from the latent feature map $\mathbf{Z}$ instead of being generated from random noise such that it is possible to reconstruct the input image. Taking inspiration from (Lin et al., 2023; Chan et al., 2023), we process $\mathbf{Z}$ with a combination of transformer blocks to facilitate learning global features and a CNN to increase the resolution of the features. For the transformer blocks after the CNN we use efficient self-attention (Xie et al., 2021) to keep the computational cost manageable for the larger resolution feature maps. We find that this combination of transformer blocks and convolutional layers achieves better novel view synthesis than using a fully convolutional architecture (see Table 4).

Next, the feature field is projected to the input view $\mathbf{P_0}$ and a novel view $\mathbf{P}_{nv}$ via volume rendering (Kajiya & Herzen, 1984; Mildenhall et al., 2020), as described in Sec. A. For the input view, we use the same fixed pose $\mathbf{P_0}$ for all instances. This means that we are modeling instances in *view space* where the coordinate system is defined from the input camera's point of view. Therefore, novel views can be sampled uniformly from a predefined range of angles around $\mathbf{P_0}$. In this work, we assume fixed camera intrinsics that we choose according to our camera settings. We find that using the same intrinsics for all datasets works well in practice (for details, see Appendix). To model unbounded scenes, we sample points along rays linearly in disparity (inverse depth) instead of depth. This effectively samples more points close to the camera and uses fewer samples at large depths. Recall that these points are projected onto triplanes for inferring the triplane features. To ensure that the full depth range is mapped onto the triplanes, we use a contraction function as in (Barron et al., 2022). The contraction function maps all coordinates to a bounded range, which ensures that sampling points are mapped to valid coordinates on the triplanes (Supp. Mat. for details). We find that representing unbounded scenes with a combination of disparity sampling and a contraction function improves novel view synthesis, see Table 4. We render low-resolution images $\hat{\mathbf{I}}^{low}$, $\hat{\mathbf{I}}_{nv}^{low}$, depth maps $\hat{\mathbf{D}}^{low}$, $\hat{\mathbf{D}}_{nv}^{low}$ and feature maps $\mathbf{F}^{low}$, $\mathbf{F}_{nv}^{low}$ from the feature field using volume rendering at $64 \times 64$ resolution, see Eq. (4). The rendered low-resolution feature maps are then processed with a superresolution CNN module (SR CNN) that increases the spatial dimensions by $4\times$ to yield the reconstructed image $\hat{\mathbf{I}}$ and a novel view image $\hat{\mathbf{I}}_{nv}$ (see Fig. 3).

**Training Objective.** We train the autoencoder with a reconstruction loss on the input view and use an adversarial objective to supervise novel views (Mi et al., 2022; Cai et al., 2022). Similar to Rombach et al. (2021), we add a small Kullback-Leibler (KL) divergence regularization term $\mathcal{L}_{KL}$ on the latent space $\mathbf{Z}$. The reconstruction loss $\mathcal{L}_{rec}$ consists of a pixel-wise loss $\mathcal{L}_{px} = |\hat{\mathbf{I}} - \mathbf{I}|$, a perceptual loss $\mathcal{L}_{VGG}$ (Zhang et al., 2018), and depth losses $\mathcal{L}_{depth}$.

As our monocular depth estimation $\mathbf{D}$ is defined only up to scale, we first compute a scale $s$ and shift $t$ for each image by solving a least-squares criterion for $s$ and $t$, which has a closed-form solution (Eigen et al., 2014). Following Ranftl et al. (2020), we enforce consistency between rendered (2D) depth $\hat{\mathbf{D}}^{low}$ and the downsampled monocular depth $\mathbf{D}^{low}$ that was estimated on the input images: $\mathcal{L}_{depth}^{2D} = ||(s\hat{\mathbf{D}}^{low}+t)-\mathbf{D}^{low}||^2$. We further found it beneficial to directly supervise the (normalized) rendering weights $w_r^i$ of the 3D sampling points (see Eq. (4)) with the depth. Let $\mathcal{K}_r(s\mathbf{D}^{low} + t)$ denote the index set of the $k$ sampling points closest to the rescaled monocular depth along ray $r$. Then,

$$\mathcal{L}_{depth}^{3D} = \sum_r \left[ (1 - \sum_{i \in \mathcal{K}_r} w_r^i)^2 + (\sum_{i \notin \mathcal{K}_r} w_r^i)^2 \right]. \tag{1}$$

Intuitively, the loss encourages large rendering weights for the points close to the re-scaled monocular depth and small rendering weights for points farther away. Note that we regularize the sum of the weights in the neighborhood $\mathcal{K}_r$ instead of the individual weights to account for imperfections in the monocular depth. In our experiments, we use a neighborhood size of $k = 5$.

In addition to the reconstruction losses on the input view, we supervise the novel views of the input image. As per-pixel supervision is not available for novel views in our setting, we follow (Mi et al., 2022; Cai et al., 2022) and use adversarial training to supervise novel views. We use a dual discriminator (Chan et al., 2022), i.e., we upsample $\hat{\mathbf{I}}_{nv}^{low}$ and concatenate it with $\hat{\mathbf{I}}_{nv}$ as input to the discriminator as a fake pair (see Fig. 3). Similarly, $\mathbf{I}$ is first downsampled to simulate a lower resolution image, and then is upsampled back to the original resolution and concatenated with the original $\mathbf{I}$ to be used as the real pair to the discriminator. Let $\mathcal{E}_\theta$, $\mathcal{G}_\psi$ and $D_\phi$ denote encoder, decoder and discriminator with parameters $\theta$, $\psi$ and $\phi$, respectively. For brevity, we omit the upsampling and concatenation of the discriminator inputs in the adversarial objective

$$V(\mathbf{I}, \mathbf{P}_{nv}, \lambda; \theta, \psi, \phi) = f\left(-D_\phi\left(\mathcal{G}_\psi(\mathcal{E}_\theta(\mathbf{I}, \mathbf{D}), \mathbf{P}_{nv})\right)\right) + f(D_\phi(\mathbf{I})) - \lambda\|\nabla D_\phi(\mathbf{I})\|^2, \tag{2}$$

where $f(x) = -\log(1 + \exp(-x))$ and $\lambda$ controls the R1-regularizer (Mescheder et al., 2018).

We find that an additional discriminator $D_\chi^{depth}$ on the low-resolution depth maps further improves novel view synthesis. $D_\chi^{depth}$ helps to ensure the volume-rendered $\hat{\mathbf{D}}^{low}$ is realistic (see Table 4). The autoencoder and discriminators are trained with alternating gradient descent steps combining the adversarial objectives with the reconstruction and regularization terms. Implementation details in Appendix.

In conclusion, to learn a latent representation suitable not only for reconstruction but also novel view synthesis, we use both reconstruction and adversarial objectives. Note, however, that this is still fundamentally different from regular GAN-like training, which most existing works on 3D-aware image synthesis rely on. Our reconstruction losses on input views prevent mode collapse and ensure stable training. This makes our approach arguably more robust and scalable. Moreover, inputs to the decoder are not sampled from random noise, like in GANs, but correspond to image encodings.

## 3.2 LATENT DIFFUSION MODEL

The autoencoder trained as described above learns a latent representation that is **(i)** compressed, and therefore suitable for efficient training of a latent diffusion model, and simultaneously also **(ii)** 3D-aware in the sense that it enables the prediction of a triplane representation from which consistent novel views can be synthesized corresponding to different viewing directions onto the modeled scene. Consequently, once the autoencoder is trained, we fit a latent diffusion model on its 3D-aware latent space in the second training stage (see Fig. 3). To this end, we encode our training images into the latent space and train the diffusion model on the encoded data. Importantly, although the autoencoder produces a compact 3D-aware latent representation, it is structured as a spatial 2D latent feature grid, as in standard LDMs for 2D image synthesis. Therefore, we can directly follow the training procedure of regular LDMs (Rombach et al., 2021) when training the diffusion model. Eventually, this allows us to train a powerful 3D-aware generative model that can be trained and sampled efficiently in 2D latent space. Our training objective is the standard denoising score matching objective as given in Eq (3), applied in latent space.

We adopt the architecture of regular 2D image LDMs (Rombach et al., 2021) to train our model. The denoiser $\mathcal{F}_\omega$ is implemented as a 2D U-Net with residual blocks (He et al., 2016) and self-attention layers (Vaswani et al., 2017). As discussed, the autoencoder's latent distribution is regularized with a KL divergence loss (Kingma & Welling, 2014; Rezende et al., 2014; Rombach et al., 2021) (see Appendix) to be roughly aligned with the standard normal distribution. However, as we enforce a very low-weighted KL loss, the distribution can have a larger variance. We estimate the standard deviation of the latent space using a batch of encoded training data and use the resulting value to normalize the latent distribution to yield a standard deviation close to 1 before fitting the diffusion model. We use the DDIM sampler (Song et al., 2021) with 200 steps. More implementation details in the Appendix.

## 4 EXPERIMENTS

The performance of LDMs is upper-bounded by the quality of the latent space they are trained on, i.e., we cannot expect to generate better novel images than what the autoencoder achieves in terms of reconstructions. Hence, a powerful autoencoder is key to training a good generative model in the second stage. We first analyze the reconstruction quality of WildFusion's autoencoder as well as its ability to synthesize novel views (Sec. 4.1). Next, we evaluate the full WildFusion model against the state-of-the-art approaches for 3D-aware image synthesis (Sec. 4.2). We provide ablation studies in Sec. 4.3. Our videos included on the project page (https://katjaschwarz.github.io/wildfusion/) show generated 3D-aware samples with camera motions. Further results are also shown in App. E.

**Datasets.** While previous 3D-aware generative models mainly focus on aligned datasets like portrait images, we study a general setting in which a canonical camera system cannot be clearly defined. Hence, we use non-aligned datasets with complex geometry: SDIP Dogs, Elephants, Horses (Mokady et al., 2022; Yu et al., 2015) as well as class-conditional ImageNet (Deng et al., 2009).
**Baselines.** We compare against the state-of-the-art generative models for 3D-aware image synthesis, EG3D (Chan et al., 2022), 3DGP (Skorokhodov et al., 2023) and POF3D (Shi et al., 2023) as well as StyleNeRF (Gu et al., 2022). 3DGP and POF3D learn a camera distribution in canonical space and can be trained on unposed images. Since we also aim to compare to other models working in the same setting as WildFusion, i.e., in view space, we adapt EG3D so that it can be trained in view space and without camera poses (indicated as EG3D* below); see Appendix for details. We also train another variant of EG3D* where we add a depth discriminator to incorporate monocular depth information. Note that the regular version of EG3D that relies on object poses is clearly outperformed by 3DGP (as shown in their paper (Skorokhodov et al., 2023)); hence, we do not explicitly compare to it.
**Evaluation Metrics.** For the autoencoder, we measure reconstruction via learned perceptual image patch similarity (LPIPS) (Zhan et al., 2018) and quantify novel view quality with Fréchet Inception Distance (nvFID) (Heusel et al., 2017) on 1000 held-out dataset images. Following prior art, we

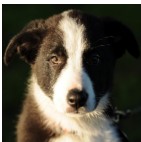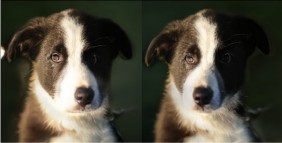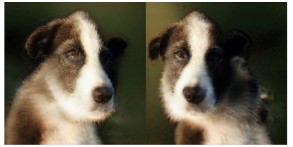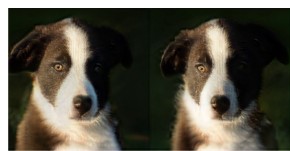

Input          WildFusion (*ours*)          EG3D*          EG3D* + $D^{depth}$

Figure 4: Baseline comparisons for novel view synthesis on images unseen during training. Shown are the input image and two novel views per method. Viewpoints across methods are the same. Included video for more results.

Table 1: Reconstruction and novel-view synthesis on SDIP Dogs, Elephants and Horses at resolution $256^2$. All evaluations on held-out test set. We report LPIPS, novel-view FID (nvFID) and non-flatness-score (NFS).

| | SDIP Dogs | | | SDIP Elephants | | | SDIP Horses | | | Rec.Time |
|---|---|---|---|---|---|---|---|---|---|---|
| | ↓LPIPS | ↓ nvFID | ↑ NFS | ↓LPIPS | ↓ nvFID | ↑ NFS | ↓LPIPS | ↓ nvFID | ↑ NFS | ↓t[s] |
| EG3D* (Chan et al., 2022) | 0.44 | 71.23 | 12.01 | 0.43 | 27.99 | 12.89 | 0.40 | 68.25 | 12.90 | > 100 |
| EG3D* + $D^{depth}$ | 0.38 | 36.65 | 14.43 | 0.45 | 56.86 | 15.74 | 0.34 | 36.27 | 12.98 | > 100 |
| WildFusion (*ours*) | **0.21** | **17.4** | **31.8** | **0.28** | **9.0** | **32.0** | **0.22** | **13.4** | **28.7** | **0.04** |

Table 2: 3D-aware image synthesis results on unimodal datasets. Baselines above double line require camera pose estimation; methods below work in view space.

| | SDIP Dogs | | | | | SDIP Elephants | | | | | SDIP Horses | | | | |
|---|---|---|---|---|---|---|---|---|---|---|---|---|---|---|---|
| | ↓FID | ↓FID$_{CLIP}$ | ↑NFS | ↑Precision | ↑Recall | ↓FID | ↓FID$_{CLIP}$ | ↑NFS | ↑Precision | ↑Recall | ↓FID | ↓FID$_{CLIP}$ | ↑NFS | ↑Precision | ↑Recall |
| POF3D (Shi et al., 2023) | 17.4 | 5.4 | 28.9 | 0.57 | 0.36 | 6.4 | 9.2 | 30.2 | 0.59 | 0.30 | 16.4 | 15.1 | **32.6** | 0.56 | 0.25 |
| 3DGP (Skorokhodov et al., 2023) | **5.9** | 6.2 | **36.3** | **0.73** | **0.38** | 3.7 | **5.9** | 32.1 | 0.67 | 0.22 | 9.0 | 13.0 | 29.2 | 0.60 | 0.28 |
| EG3D* (Chan et al., 2022) | 16.3 | 5.8 | 11.8 | 0.60 | 0.29 | 3.0 | 6.8 | 13.3 | 0.59 | 0.31 | 10.2 | 6.7 | 14.1 | 0.57 | 0.23 |
| EG3D* + $D_{depth}$ | 18.7 | 8.8 | 13.9 | 0.71 | 0.15 | 4.5 | 8.5 | 18.3 | 0.56 | 0.24 | 6.5 | 8.7 | 13.7 | 0.59 | 0.30 |
| StyleNeRF (Gu et al., 2022) | 12.3 | 7.9 | 30.0 | 0.65 | 0.34 | 10.0 | 9.1 | 20.1 | 0.53 | 0.17 | 4.5 | **8.1** | 27.2 | 0.65 | 0.35 |
| WildFusion (*ours*) | 12.2 | **5.2** | 31.7 | 0.66 | **0.38** | **2.9** | 6.5 | **32.2** | **0.70** | **0.34** | **4.3** | 8.8 | 28.8 | **0.70** | **0.37** |

sample camera poses around the input view $\mathbf{P}_0$ from Gaussian distributions with $\sigma = 0.3$ and 0.15 radians for the yaw and pitch angles (Chan et al., 2021). We also report non-flatness-score (NFS) (Skorokhodov et al., 2023). It measures average entropy of the normalized depth maps' histograms and quantifies surface flatness, indicating geometry quality. For the full generative models, we measure NFS and evaluate FID between 10k generated images and the full dataset, sampling camera poses as for nvFID. As FID can be prone to distortions (Kynkäänniemi et al., 2023), we also show FID$_{CLIP}$, which uses CLIP features. To quantify diversity, we report Recall, and we also show Precision (Sajjadi et al., 2018; Kynkäänniemi et al., 2019). Qualitative results display a view range of $30°$ and $15°$ for azimuth and polar angles around the input view, similar to Skorokhodov et al. (2023).

## 4.1 AUTOENCODER FOR RECONSTRUCTION AND NOVEL-VIEW SYNTHESIS

As EG3D (Chan et al., 2022) is a GAN-based approach, we need to perform GAN-inversion (Chan et al., 2022; Roich et al., 2023) to reconstruct input images (we use the scripts provided by the authors). Quantitative results are in Table 1, qualitative comparisons in Fig. 4 (more in App. E). Compared to EG3D using GAN-inversion, WildFusion's autoencoder achieves superior performance on all metrics and is also more efficient, since we do not require a lengthy optimization process (which occasionally diverges) to embed input images into latent space. Despite its low latent space dimension of $32 \times 32 \times 4$, our autoencoder achieves both good compression and novel view synthesis. In Fig. 1, Fig. 5 and in App. E, we also show novel view synthesis results on ImageNet, which performs equally well.

## 4.2 3D-AWARE IMAGE SYNTHESIS WITH LATENT DIFFUSION MODELS

**SDIP Datasets.** We compare WildFusion against state-of-the-art 3D-aware generative models (Table 2) and provide model samples in the App. E. Our videos in the Supp. Mat. show more generated 3D-aware samples, including smooth viewpoint changes. Compared to EG3D* (Chan et al., 2022), which effectively also models instances in view-space, WildFusion achieves higher performance on all metrics, i.e., in terms of image quality (FID/FID$_{CLIP}$), 3D geometry (NFS), and diversity (Recall). This validates the effectiveness of our LDM framework in this setting. When also adding a depth discriminator to EG3D*, WildFusion still achieves better FID and NFS and Recall. Note that in particular on NFS and Recall, the performance gap is generally large. We also outperform StyleNeRF. We conclude that previous works that operate in view space can struggle with flat geometry and distribution coverage when training on reasonably complex, non-aligned images. In contrast, WildFusion achieves state-of-the-art performance in this setting.

The baselines 3DGP (Skorokhodov et al., 2023) and POF3D (Shi et al., 2023) both rely on sophisticated camera pose estimation procedures and do not operate in view space, which makes a direct

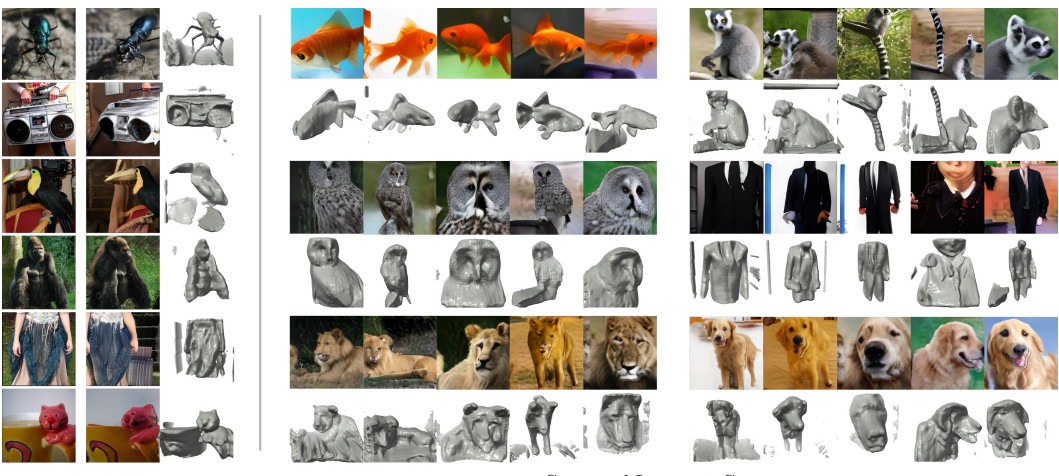

Input  Novel View + Geom.                    Generated Images + Geometry

Figure 5: Generated 3D-aware image samples and geometry by WildFusion. Included videos for more results.

|  | ↓FID | ↓FID$_{\text{CLIP}}$ | ↑NFS | ↑Precision | ↑Recall |
|---|---|---|---|---|---|
| 3DGP (Skorokhodov et al., 2023) | **19.7** | **8.1** | 18.5 | 0.31 | 0.02 |
| EG3D* + $D_{\text{depth}}$ | 111.6 | 27.2 | 20.6 | 0.48 | 0.01 |
| StyleNeRF (Gu et al., 2022) | 77.7 | 17.5 | 28.0 | 0.38 | 0.03 |
| WildFusion (*ours*), $s = 1$ | 65.1 | 15.3 | 33.6 | 0.58 | 0.16 |
| WildFusion (*ours*), $s = 2$ | 35.4 | 11.7 | **33.8** | **0.59** | **0.20** |
| WildFusion (*ours*), $s = 5$ | 25.5 | 11.6 | 33.6 | 0.53 | 0.13 |

Table 3: 3D-aware image synthesis results on class-conditional ImageNet. For WildFusion, we report results for different classifier-free guidance scales $s$ (App. C.2); extended results in Table 7.

comparison with WildFusion difficult. Nevertheless, WildFusion performs on-par or even better on the different metrics, despite not relying on posed images or learned pose or camera distributions at all. These previous works' reliance on complex canonical camera systems represents a major limitation with regards to their scalability, which our 3D-aware LDM framework avoids. Note that in particular on Recall, WildFusion always performs superior compared to all other methods, demonstrating that diffusion-based frameworks are usually better than GAN-based ones in terms of sample diversity.

**ImageNet.** We find that WildFusion outperforms all baselines on NFS, Precision and Recall by large margins, for varying classifier-free guidance scales (Table 3). The extremely low Recall scores of the GAN-based baselines indicate very low sample diversity (mode collapse). We visually validate this for 3DGP, the strongest of the three baselines, in Fig. 2: 3DGP collapses and produces almost identical samples within classes, showing virtually no diversity. In contrast, our model produces diverse, high-quality samples. Note that this failure of 3DGP is pointed out by the authors (see *"Limitations and failure cases"* at https://snap-research.github.io/3dgp/). The FID metric does not accurately capture that mode collapse. While we outperform EG3D and StyleNeRF also on FID, WildFusion is slightly behind 3DGP. However, it is known that FID is a questionable metric (Kynkäänniemi et al., 2023) and the qualitative results in Fig. 5 and Fig. 2 show that WildFusion generates diverse and high-fidelity images and generally learns a reasonable geometry in this challenging setting. We believe that a mode-collapsed generative model like 3DGP, despite good FID scores, is not useful in practice. Finally, note that due to limited compute resources our ImageNet model was trained only with a total batch size of 256. However, non-3D-aware ImageNet diffusion models are typically trained on batch sizes >1000 to achieve strong performance (Rombach et al., 2021; Karras et al., 2022; Kingma & Gao, 2023). Hence, scaling our model and the compute resources would likely boost the results significantly.

**Interpolation and Generative Resampling.** In Fig. 6, we use WildFusion to perform semantically meaningful 3D-aware image interpolation between two given (or generated) images. Moreover, in Fig. 7 we demonstrate how we can use our 3D-aware latent diffusion model to refine images and geometry by only partially diffusing their encodings and regenerating from those intermediate diffusion levels. These two applications highlight the versatility of WildFusion and have potential use cases in 3D-aware image editing. To the best of our knowledge, this is the first time such applications have been demonstrated for such 3D-aware image generative models. See the project

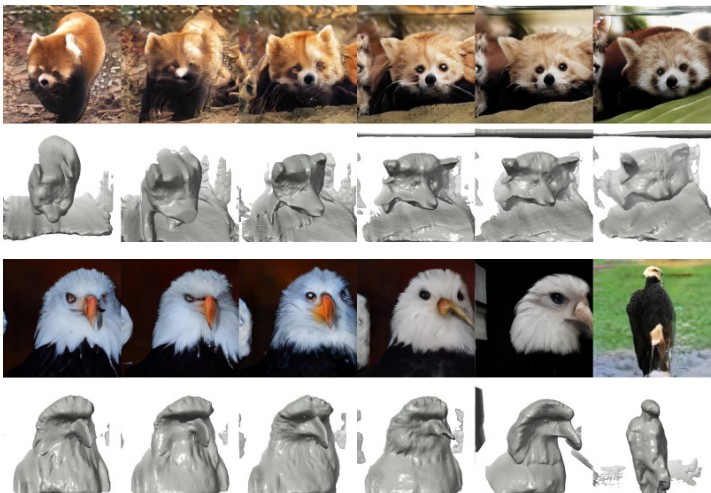

Figure 6: **3D-Aware Image Interpolation.** We encode two images into latent space (far left and far right), further encode into the diffusion model's Gaussian prior space (inverse DDIM), interpolate the resulting encodings, and generate the corresponding 3D images along the interpolation path.

Figure 7: **3D-Aware Generative Image Resampling.** Given an image (far left), we forward diffuse its latent encoding for varying numbers of steps and re-generate from the partially noised encodings. Depending on how far we diffuse, we obtain varying levels of generative image resampling.

page (https://katjaschwarz.github.io/wildfusion/) for animations with viewpoint changes for these 3D-aware image interpolation and generative image resampling results.

### 4.3 ABLATION STUDIES

We provide a detailed ablation study in Table 4, starting from a base model, where the decoder architecture follows EG3D's generator. We gradually introduce changes and track their impact on novel view synthesis (nvFID) and geometry (NFS) (corresponding samples in App. E in Fig. 12). For computational efficiency, we perform the study at image resolution $128^2$ and reduce the number of network parameters compared to our main models. Like LDMs (Rombach et al., 2021), the initial setting is trained only with reconstruction losses. Unsurprisingly, this results in planar geometry, indicated by low NFS. This changes when a discriminator on novel views is added. However, geometry becomes noisy and incomplete, indicated by high NFS and worse nvFID. We hypothesize the purely convolutional architecture is suboptimal. Hence, in the next step, we instead use a combination of convolutions and transformer blocks (ViT (Dosovitskiy et al., 2020)) in the decoder, which improves novel view quality and results in less noisy geometry.

Adding the monocular depth discriminator $D_\chi^{depth}$ significantly improves nvFID, and we observe an even bigger improvement when tailoring the model to represent unbounded scenes with disparity sampling and a coordinate contraction function. Further supervision on the rendered depth ($\mathcal{L}_{depth}^{2D}$) does not improve results but as it does not hurt performance either, we kept it in our pipeline. Lastly, we find that both adding depth as an input to the encoder and directly supervising the rendering weights with $\mathcal{L}_{depth}^{3D}$ result in slight improvements in NFS and qualitatively improve geometry.

Table 4: Ablation study on SDIP Dogs.

| Model configuration | ↓ nvFID | ↑NFS |
|---|---|---|
| Base config | 53.2 | 10.3 |
| + $D_\phi$ | 61.1 | 37.0 |
| + ViT backbone | 48.3 | 33.9 |
| + $D_\chi^{depth}$ | 40.5 | 34.0 |
| + modeling unbounded scenes | 32.8 | 32.4 |
| + $\mathcal{L}_{depth}^{2D}$ | 34.6 | 32.0 |
| + encode depth | 33.3 | 33.3 |
| + $\mathcal{L}_{depth}^{3D}$ | 34.0 | 33.7 |

## 5 CONCLUSIONS

We introduce WildFusion, a 3D-aware LDM for 3D-aware image synthesis. WildFusion is trained without multiview or 3D geometry supervision and relies neither on posed images nor on learned pose or camera distributions. Key to our framework is an image autoencoder with a 3D-aware latent space that simultaneously enables not only novel view synthesis but also compression. This allows us to efficiently train a diffusion model in the autoencoder's latent space. WildFusion outperforms recent state-of-the-art GAN-based methods when training on diverse data without camera poses. Future work could scale up 3D-aware LDMs to the text-conditional setting, similar to how 2D diffusion models have been applied on extremely diverse datasets (Ramesh et al., 2022; Rombach et al., 2021; Saharia et al., 2022; Balaji et al., 2022).

## ACKNOWLEDGMENTS

Katja Schwarz and Andreas Geiger were supported by the ERC Starting Grant LEGO-3D (850533) and the DFG EXC number 2064/1 - project number 390727645. The authors thank the International Max Planck Research School for Intelligent Systems (IMPRS-IS) for supporting Katja Schwarz. Lastly, we would like to thank Nicolas Guenther for his general support.

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

# A  THEORETICAL BACKGROUND

*We summarize the most relevant theoretical concepts to our work in the following.*

**Diffusion Models** (Sohl-Dickstein et al., 2015; Ho et al., 2020; Song et al., 2020) create diffused inputs $\mathbf{x}_\tau = \alpha_\tau \mathbf{x} + \sigma_\tau \boldsymbol{\epsilon}$, $\boldsymbol{\epsilon} \sim \mathcal{N}(\mathbf{0}, \mathbf{I})$ from data $\mathbf{x} \sim p_{\text{data}}$, where $\alpha_\tau$ and $\sigma_\tau$ define a noise schedule, parameterized by a diffusion-time $\tau$. A denoiser model $\mathcal{F}_\omega$ with parameters $\omega$ is trained to denoise the perturbed data via denoising score matching (Hyvärinen, 2005; Vincent, 2011; Song & Ermon, 2019),

$$\arg\min_\omega \ \mathbb{E}_{\mathbf{x} \sim p_{\text{data}}, \tau \sim p_\tau, \boldsymbol{\epsilon} \sim \mathcal{N}(\mathbf{0}, \mathbf{I})} \left[ \| \mathbf{v} - \mathcal{F}_\omega(\mathbf{x}_\tau, \tau) \|_2^2 \right], \tag{3}$$

with the target $\mathbf{v} = \alpha_\tau \boldsymbol{\epsilon} - \sigma_\tau \mathbf{x}$ (this is known as **v**-*prediction* (Salimans & Ho, 2022)). Further, $p_\tau$ is a uniform distribution over the diffusion time $\tau$, such that the model is trained to denoise for all different times $\tau$. The noise schedule is designed such that input data is entirely perturbed into Gaussian random noise after the maximum diffusion time. An iterative generative denoising process that employs the learned denoiser $\mathcal{F}_\omega$ can then be initialized from such Gaussian noise to synthesize novel data. Classifier-free guidance can be used to amplify conditioning strength when conditioning the diffusion model on data such as classes; see Ho & Salimans (2021) and App. C.2.

Diffusion models have also been applied to 3D data (Zeng et al., 2022; Wang et al., 2022b; Bautista et al., 2022; Shue et al., 2022; Nam et al., 2022) but usually require explicit 3D or multiview super-vision. In contrast,WildFusion learns from an unstructured image set without multiview supervision.

**Latent Diffusion Models (LDMs)** (Rombach et al., 2021; Vahdat et al., 2021) first train a regularized autoencoder with encoder $\mathcal{E}$ and decoder $\mathcal{D}$ to transform input images $\mathbf{I} \sim p_{\text{data}}$ into a spatially lower-dimensional latent space $\mathbf{Z}$ of reduced complexity, from which the original data can be reconstructed, this is, $\hat{\mathbf{I}} = \mathcal{D}(\mathcal{E}(\mathbf{I})) \approx \mathbf{I}$. A diffusion model is then trained in the compressed latent space, with $\mathbf{x}$ in Eq. (3) replaced by an image's latent representation $\mathbf{Z} = \mathcal{E}(\mathbf{I})$. This latent space diffusion model can be typically smaller in terms of parameter count and memory consumption compared to corresponding pixel-space diffusion models of similar performance. More diffusion model details in Appendix.

**3D-Representations for 3D-Aware Image Synthesis.** 3D-aware generative models typically generate neural radiance fields or feature fields, i.e., they represent a scene by generating a color or a feature value $\mathbf{f}$ and a density $\sigma$ for each 3D point $\mathbf{p} \in \mathbb{R}^3$ (Mildenhall et al., 2020; Schwarz et al., 2020; Niemeyer & Geiger, 2021a). Features and densities can be efficiently computed from a triplane representation $[\mathbf{T}_{xy}, \mathbf{T}_{xz}, \mathbf{T}_{yz}]$ (Chan et al., 2022; Peng et al., 2020). The triplane feature $\mathbf{t}$ is obtained by projecting $\mathbf{p}$ onto each of the three feature planes and averaging their feature vectors $(\mathbf{t}_{xy}, \mathbf{t}_{xz}, \mathbf{t}_{yz})$. An MLP then converts the triplane feature $\mathbf{t}$ to a feature and density value $[\mathbf{f}, \sigma] = MLP(\mathbf{t})$.

Given a camera pose, the feature field is rendered via volume rendering (Kajiya & Herzen, 1984; Mildenhall et al., 2020). For that, the feature field is evaluated at discrete points $\mathbf{p}_r^i$ along each camera ray $r$ yielding features and densities $\{(\mathbf{f}_r^i, \sigma_r^i)\}_{i=1}^N$. For each ray $r$, these features are aggregated to a feature $\mathbf{f}_r$ using alpha composition

$$\mathbf{f}_r = \sum_{i=1}^N w_r^i \mathbf{f}_r^i, \qquad w_r^i = T_r^i \alpha_r^i, \qquad T_r^i = \prod_{j=1}^{i-1} \left( 1 - \alpha_r^j \right), \qquad \alpha_r^i = 1 - \exp\left( -\sigma_r^i \delta_r^i \right), \tag{4}$$

where $T_r^i$ and $\alpha_r^i$ denote the transmittance and alpha value of sample point $\mathbf{p}_r^i$ along ray $r$ and $\delta_r^i = \left\| \mathbf{p}_r^{i+1} - \mathbf{p}_r^i \right\|_2$ is the distance between neighboring sample points. Similarly, depth can be rendered, see Appendix. For efficiency, a low-resolution feature map, and optionally a low-resolution image $\hat{\mathbf{I}}^{low}$, can be rendered instead of an image at full resolution (Niemeyer & Geiger, 2021a; Chan et al., 2022). The feature map is then subsequently upsampled and decoded into a higher-resolution image $\hat{\mathbf{I}}$.

# B  RELATED WORK

*Here, we present an extended discussion about related work.*

**Diffusion Models.** Diffusion models (DMs) (Sohl-Dickstein et al., 2015; Ho et al., 2020; Song et al., 2020) have proven to be powerful image generators, yielding state-of-the art results in unconditional as well as class- and text-guided synthesis (Nichol & Dhariwal, 2021; Rombach et al., 2021; Dhariwal

& Nichol, 2021a; Ho et al., 2022; Dockhorn et al., 2022a;b; Vahdat et al., 2021; Nichol et al., 2022b; Ramesh et al., 2022; Saharia et al., 2022; Balaji et al., 2022). However, none of these works tackles 3D-aware image synthesis.

**3D Diffusion Models.** There is also much literature on applying diffusion models to 3D data, e.g. 3D point clouds (Zhou et al., 2021a; Luo & Hu, 2021; Zeng et al., 2022) or tetrahedral meshes (Kalischek et al., 2022). Shue et al. (2022) learn a diffusion model on a triplane representation parameterizing a neural occupancy field. GAUDI (Bautista et al., 2022) and 3D-LDM (Nam et al., 2022) train diffusion models in latent spaces learnt using an autodecoder framework and generate 3D scenes and 3D shapes, respectively. RODIN (Wang et al., 2022b) proposes a hierarchical latent diffusion model framework to learn 3D human avatars and NF-LDM (Kim et al., 2023) trains a hierarchical diffusion model for outdoor scene generation. Dupont et al. (2022) and Du et al. (2021) treat data as functions and also explore encoding 3D signals into latent spaces, but using more inefficient meta-learning (Dupont et al., 2022) or auto-decoder (Du et al., 2021) methods. Dupont et al. (2022) also trains a diffusion model on the encoded 3D data. However, all aforementioned works rely on explicit 3D or multiview supervision. In contrast, our approach learns from an unstructured image collection without multiview supervision.

RenderDiffusion (Anciukevicius et al., 2023) trains a diffusion model directly on images, using a triplanar 3D feature representation inside the denoiser network architecture, thereby enabling 3D-aware image generation during synthesis. However, scaling RenderDiffusion to high-resolution outputs is challenging, as it operates directly on images. In fact, it considers only small image resolutions of 32x32 or 64x64, likely due to computational limitations. When trained on single-view real-world data, the paper only considers data with little pose variation (human and cat faces) and it is unclear whether the approach is scalable to diverse or high-resolution image data (moreover, no perceptual quality evaluations on metrics such as FID are presented and there is no code available). In contrast, our diffusion model is trained efficiently in a low-resolution, compressed and 3D-aware latent space, while simultaneously predicting high-resolution triplanes and enabling high-resolution rendering. Hence, WildFusion generates significantly higher quality 3D-aware images than RenderDiffusion and it is scalable to diverse datasets such as ImageNet, as we demonstrate.

Concurrently with us, IVID (Xiang et al., 2023b) trains a 2D diffusion model that first synthesizes an initial image and then iteratively generates novel views conditioned on it. However, the iterative generation is extremely slow because it requires running the full reverse diffusion process for every novel view. Further, an explicit 3D representation can only be constructed indirectly from a large collection of generated multi-view images, afterwards. Instead, WildFusion uses a fundamentally different approach and only runs the reverse diffusion process once to generate a (latent) 3D representation from which multi-view images can be rendered directly and geometry can be extracted easily. At time of submission code for experimental comparisons to IVID was not available.

**Optimization from Text-to-Image Diffusion Models.** Another line of work distills 3D objects from large-scale 2D text-to-image diffusion models (Poole et al., 2022; Lin et al., 2022; Nichol et al., 2022a; Metzer et al., 2022; Wang et al., 2022a; Deng et al., 2022a). However, these methods follow an entirely different approach compared to 3D- and 3D-aware diffusion models and require a slow optimization process that needs to be run per instance.

**3D-Aware Image Synthesis.** 3D-aware generative models consider image synthesis with control over the camera viewpoint (Liao et al., 2020; Schwarz et al., 2020; Chan et al., 2021). Most existing works rely on generative adversarial networks (GANs) (Goodfellow et al., 2014) and use coordinate-based MLPs as 3D-generator (Schwarz et al., 2020; Chan et al., 2021; Gu et al., 2022; Jo et al., 2021; Xu et al., 2022; Zhou et al., 2021b; Zhang et al., 2021; Or-El et al., 2022; Xu et al., 2021; Pan et al., 2021; Deng et al., 2022b; Xiang et al., 2023a; Gu et al., 2022), building on Neural Radiance Fields (Mildenhall et al., 2020) as 3D representation. VoxGRAF (Schwarz et al., 2022) and EG3D (Chan et al., 2022) proposed efficient convolutional 3D generators that require only a single forward pass for generating a 3D scene. Our autoencoder uses EG3D's triplane representation and their dual discriminator to improve view consistency. Early 3D-aware generative models that do not require camera poses during training and operate in view space include HoloGAN Nguyen-Phuoc et al. (2019) and PlatonicGAN (Henzler et al., 2019). They are outperformed, for instance, by the more recent StyleNeRF (Gu et al., 2022), which uses a style-based architecture (Karras et al., 2019;

2020) and proposes a novel path regularization loss to achieve 3D consistency. In contrast to the aforementioned approaches, however, our generative model is not a GAN. GANs are notoriously hard to train (Mescheder et al., 2018) and often do not cover the data distribution well. Instead, we explore 3D-aware image synthesis with latent diffusion models for the first time.

Until recently, 3D-aware image synthesis focused on aligned datasets with well-defined pose distributions, such as portrait images (Liu et al., 2015; Karras et al., 2019). For instance, POF3D (Shi et al., 2023) is a recent 3D-aware GAN that infers camera poses and works in a canonical view space; it has been used only for datasets with simple pose distributions, such as cat and human faces. Advancing to more complex datasets, GINA-3D learns to generate assets from driving data, assuming known camera and LIDAR sensors. The two-stage approach first trains a vision transformer encoder yielding a latent triplane representation. Next, a discrete token generative model is trained in the latent space. We consider a setting where camera information and ground truth depth are not available and train a latent diffusion model on 2D feature maps.

To scale 3D-aware image synthesis to more complex datasets, i.e. ImageNet, 3DGP (Skorokhodov et al., 2023) proposes an elaborate camera model and learns to refine an initial prior on the pose distribution. Specifically, 3DGP predicts the camera location in a canonical coordinate system per class and sample-specific camera rotation and intrinsics. This assumes that samples within a class share a canonical system, and we observe that learning this complex distribution can aggravate training instability. Further, the approach needs to be trained on heavily filtered training data. In contrast, WildFusion can generate high-quality and diverse samples even when trained on the entire ImageNet dataset without any filtering (see Sec. 4.2).

Concurrently to WildFusion, VQ3D (Sargent et al., 2023) proposes an autoencoder architecture, but uses sequence-like latent variables and trains an autoregressive transformer in the latent space. Instead, WildFusion trains a diffusion model on latent feature maps. Another difference is that VQ3D applies two discriminators on the generated images, one that distinguishes between reconstruction and training image, and another one that discriminates between reconstruction and novel view. WildFusion only applies a single discriminator to supervise the novel views and instead has an additional discriminator on the depth. At time of submission code for experimental comparisons was not available.

**Novel View Synthesis.** Our autoencoder is related to methods that synthesize novel views given a single input image: LoLNeRF (Rebain et al., 2021) trains an autodecoder with per-pixel reconstruction loss and mask supervision. In contrast, we add an adversarial objective to supervise novel views. Mi et al. (2022) proposes a similar approach but is not investigated in the context of generative modeling. Another line of recent works leverage GAN inversion to generate novel views from single input images (Li et al., 2022; Cai et al., 2022; Lin et al., 2023; Xie et al., 2022; Yin et al., 2022; Lan et al., 2022; Pavllo et al., 2022) but rely on pretrained 3D-aware GANs and thereby inherit their aforementioned limitations. Several recent works (Watson et al., 2022; Chan et al., 2023; Liu et al., 2023) tackled novel view synthesis with view-conditioned 2D diffusion models, but are trained with explicit 3D or multiview supervision. Unlike these approaches, we use our autoencoder to also train a 3D-aware generative model.

## C   IMPLEMENTATION DETAILS

### C.1   CAMERA SYSTEM

In the following, we describe the camera system we use to learn a 3D representation in view space. An overview is shown in Fig. 8. The input view $\mathbf{P}_0$ is defined by the camera intrinsics $\mathbf{K}_0$ and extrinsics $[\mathbf{R}_0, \mathbf{T}_0]$, where $\mathbf{R}_0$ and $\mathbf{T}_0$ denote the rotation and translation of the camera in the world coordinate system. We fix $\mathbf{R}_0$ and $\mathbf{T}_0$, such that the camera is located at a fixed radius and looks at the origin of the world coordinate center. For the intrinsics $\mathbf{K}_0$, we choose a small, fixed field of view since most images in the datasets are cropped and perspective distortion is generally small. For the experiments on all datasets, we set

$$\mathbf{R}_0 = \begin{pmatrix} 1 & 0 & 0 \\ 0 & -1 & 0 \\ 0 & 0 & -1 \end{pmatrix} \quad \mathbf{T}_0 = \begin{pmatrix} 0 \\ 0 \\ 2.7 \end{pmatrix} \quad \mathbf{K}_0 = \begin{pmatrix} 5.4 & 0 & 0.5 \\ 0 & 5.4 & 0.5 \\ 0 & 0 & 1.0 \end{pmatrix} . \tag{5}$$

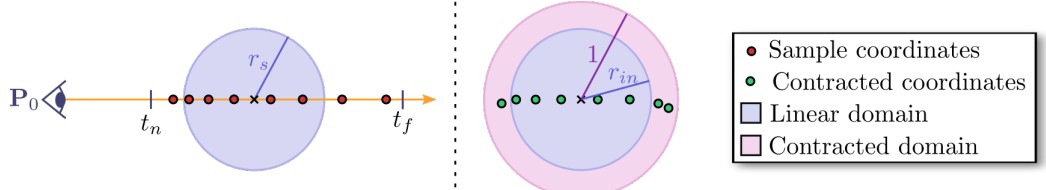

Figure 8: **Camera System:** Our autoencoder models objects in view space, i.e., it uses a fixed camera pose $P_0$ to render the reconstructed images. The orange line represents a camera ray and $t_n$ and $t_f$ denote the near and the far plane, respectively, between which the 3D object is located. We evaluate the camera ray at the red sample coordinates which are spaced linearly in disparity (inverse depth) to better model large depth ranges. However, the triplanes that carry the features for encoding the 3D object are only defined within a normalized range ($[-1, 1]$ in all directions); hence, we need to normalize the samples on the camera ray accordingly to ensure that all coordinates are projected onto the triplanes. Specifically, we use the contraction function in Eq. (6). Sample coordinates within a fixed radius $r$ are mapped linearly to a sphere of radius $r_{in} < 1$, i.e., into the domain where the triplanes are defined (linear domain). Sample coordinates with norm $> r$ are contracted such that they have norm $\leq 1$ in the domain of the triplanes (contracted domain).

During training, we sample novel views around $\mathbf{P}_0$ by drawing offsets for azimuth and polar angles uniformly from $[-35°, 35°]$ and $[-15°, 15°]$, respectively.

As we consider unbounded scenes, we sample points along the camera rays linearly in disparity (inverse depth). Thereby, we effectively sample more points close to the camera and use fewer samples at large depths where fewer details need to be modeled. In practice, we set the near and far planes to $t_n = 2.25$ and $t_f = 5.0$, respectively. During rendering, the sampled points are mapped to 3D coordinates in $[-1, 1]$ and subsequently projected to 2D points on the triplanes (see Fig. 8). Reflecting the disparity sampling, we choose a non-linear mapping function for the 3D coordinates that places sampling points with a large depth closer together. This assigns a larger area on the triplanes to points close to the camera while points far away are projected onto a smaller area. Specifically, we use a contraction function inspired by Barron et al. (2022) that maps points within a sphere of radius $r_s$ to a sphere of radius $r_{in} < 1$ and all points outside of the sphere to a sphere of radius 1. Let $\mathbf{x} \in \mathbb{R}^3$ denote a sampled point; then, the contracted coordinate $\mathbf{x}_c$ is calculated as

$$
\mathbf{x}_c = \begin{cases} \mathbf{x}\frac{r_{in}}{r}, & \text{if } ||\mathbf{x}|| \leq r \\ \left( (1 - r_{in}) \left( 1 - \frac{1}{||\mathbf{x}||-r+1} \right) + r_{in} \right) \frac{\mathbf{x}}{||\mathbf{x}||}, & \text{otherwise} \end{cases}
\tag{6}
$$

We set $r_s = 1.3$ and $r_{in} = 0.8$ for all experiments.

## C.2 Network Architecture, Objectives, and Training

**First Stage: 3D-aware Autoencoder.** The encoder network is a feature pyramid network (FPN) (Lin et al., 2017). In practice, we use the setup from variational autoencoders (VAEs) (Kingma & Welling, 2014; Rezende et al., 2014) and predict means $\boldsymbol{\mu}$ and variances $\boldsymbol{\sigma}^2$ of a normal distribution from which we sample the latent representation $\mathbf{Z}$

$$
[\boldsymbol{\mu}, \boldsymbol{\sigma}^2] = FPN(\mathbf{I}), \qquad \mathbf{Z} \sim \mathcal{N}(\boldsymbol{\mu}, \boldsymbol{\sigma}^2),
\tag{7}
$$

where $\mathbf{Z} \in \mathbb{R}^{c \times h \times w}$ (formally, we assume a diagonal covariance matrix and predict means and variances for all latent dimensions independently). We regularize the latent space by minimizing a low-weighted Kullback-Leibler divergence loss $\mathcal{L}_{KL}$ between $q_{\mathcal{E}}(\mathbf{Z}|\mathbf{I}_P) = \mathcal{N}(\boldsymbol{\mu}, \boldsymbol{\sigma}^2)$ and a standard normal distribution $\mathcal{N}(\mathbf{0}, \mathbf{I})$.

The decoder consists of transformer blocks at resolution of the feature map $\mathbf{Z}$ followed by a CNN that increases the resolution to $128^2$ pixels. The CNN applies grouped convolutions with three groups to prevent undesired spatial correlation between the triplanes, see Wang et al. (2022b). At resolution $128^2$ pixels, we then add two further transformer blocks with efficient self-attention (Xie et al., 2021) to facilitate learning the correct spatial correlations for the triplanes. The triplanes have a resolution of $128^2$ pixels. We use the superresolution modules and dual discriminator from EG3D (Chan et al., 2022). The main task of the discriminator is to provide supervision for novel views, but it can also be used to improve the details in the reconstructed views (Esser et al., 2021). We hence use $95\%$ novel views and $5\%$ input views when training the discriminator. For the main models in the paper, the encoder has $\sim 32M$ parameters. We use 8 transformer blocks in the decoder accumulating to $\sim 26M$

Table 5: Weights of all losses used in the training objective.

|  | $\lambda_{px}$ | $\lambda_{VGG}$ | $\lambda_{depth}^{2D}$ | $\lambda_{depth}^{3D}$ | $\lambda_{KL}$ | $\lambda$ | $\lambda_d$ |
|---|---|---|---|---|---|---|---|
| Weight | 10 | 10 | 1 | 1 | 1e-4 | 1 | 10 |

Table 6: Hyperparameters for our latent diffusion models.

| Architecture | | Training | | Diffusion Setup | |
|---|---|---|---|---|---|
| Image shape | $256 \times 256 \times 3$ | Parameterization | $v$ | Diffusion steps | 1000 |
| $z$-shape | $32 \times 32 \times 4$ | Learning rate | $10^{-4}$ | Noise schedule | Cosine |
| Channels | 224 | Batch size per GPU | 64 | Offset $s$ | 0.008 |
| Depth | 2 | #GPUs | 4 | Scale factor $z$ | 0.5 |
| Channel multiplier | 1,2,4,4 | $p_{\mathrm{drop}}$ | 0.1 | Sampler | DDIM |
| Attention resolutions | 32,16,8 | | | Steps | 200 |
| Head channels | 32 | | | $\eta$ | 1.0 |

parameters for the full decoder. The discriminator has $\sim 29M$ parameters. For computational efficiency, ablation studies (Table 3 of the main paper) are performed with downscaled models and at an image resolution of $128^2$ pixels instead of $256^2$ pixels. The downscaled models have a reduced channel dimension; specifically, the triplane resolution is reduced to $64^2$ pixels and the decoder uses 4 instead of 8 transformer blocks. The resulting models count $1.6M$, $2.5M$, and $1.8M$ parameters for encoder, decoder and discriminator, respectively.

The autoencoder uses Adam (Kingma & Ba, 2015) with a learning rate of $1.4 \times 10^{-4}$. However, for the superresolution modules and dual discriminator, we found it important to use the original learning rates from EG3D, which are $2 \times 10^{-3}$ and $1.9 \times 10^{-3}$, respectively. We train all autoencoders with a batch size of 32 on 8 NVIDIA A100-PCIE-40GB GPUs until the discriminator has seen around $5.5M$ training images. Training our autoencoder in this setting takes around 2.5 days.

**Implementation and Training.** Our autoencoder is trained with a reconstruction loss on the input view and an adversarial objective to supervise novel views (Mi et al., 2022; Cai et al., 2022). Similar to Rombach et al. (2021), we add a small Kullback-Leibler (KL) divergence regularization term $\mathcal{L}_{KL}$ on the latent space $\mathbf{Z}$, as discussed above. The reconstruction loss $\mathcal{L}_{rec}$ consists of a pixel-wise loss $\mathcal{L}_{px} = |\hat{\mathbf{I}} - \mathbf{I}|$, a perceptual loss $\mathcal{L}_{VGG}$ (Zhang et al., 2018), and depth losses $\mathcal{L}_{depth}^{2D}$, $\mathcal{L}_{depth}^{3D}$. The full training objective is as follows

$$\mathcal{L}_{rec} = \lambda_{px}\mathcal{L}_{px} + \lambda_{VGG}\mathcal{L}_{VGG} + \lambda_{depth}^{2D}\mathcal{L}_{depth}^{2D} + \lambda_{depth}^{3D}\mathcal{L}_{depth}^{3D} \tag{8}$$

$$\min_{\theta,\psi} \max_{\phi} [V(\mathbf{I}, \mathbf{P}_{nv}, \lambda; \theta, \psi, \phi) + V_{depth}(\mathbf{I}, \mathbf{P}_{nv}, \lambda_d; \theta, \psi, \chi)$$

$$+ \mathcal{L}_{rec}(\mathbf{I}_P, \mathbf{P}; \theta, \psi) + \lambda_{KL}\mathcal{L}_{KL}(\mathbf{I}_P; \theta)] \tag{9}$$

where $\lambda_{\{\}}$ weigh the individual loss terms ($\lambda$ without subscript denotes the R1 regularization coefficient for the regular discriminator and $\lambda_d$ is the R1 regularization coefficient for the depth discriminator). The values for the weights are summarized in Table 5. Note that $V$ and $V_{depth}$ denote the adversarial objectives of the regular and depth discriminator, respectively (see Eq. (5) in main paper).

Our code base builds on the official PyTorch implementation of StyleGAN (Karras et al., 2019) available at https://github.com/NVlabs/stylegan3, EG3D (Chan et al., 2022) available at https://github.com/NVlabs/eg3d and LDM (Rombach et al., 2021) available at https://github.com/CompVis/latent-diffusion. Similar to StyleGAN, we use a minibatch standard deviation layer at the end of the discriminator (Karras et al., 2018) and apply an exponential moving average of the autoencoder weights. Unlike Karras et al. (2019), we do not train with path regularization or style-mixing. To reduce computational cost and overall memory usage R1-regularization (Mescheder et al., 2018) is applied only once every 16 minibatches (also see R1-regularization coefficients $\lambda$ and $\lambda_d$ in Table 5).

**Second Stage: Latent Diffusion Model.** We provide detailed model and training hyperparameter choices in Table 6. We follow the naming convention from LDM Rombach et al. (2021) and train the

models for $\sim$200 epochs for each dataset. Our LDMs are trained on 4 NVIDIA A100-PCIE-40GB GPUs for 8 hours on SDIP elephant and for 1 day on SDIP horse, dog. On ImageNet, we train a class-conditional model for $\sim$5 days, doubling the batch size from 128 to 256. Otherwise, we use the same hyperparameters and model size due to computational constraints.

**Guidance.** Class-conditioning is implemented through cross attention with learned embeddings, following Rombach et al. (2021). We also drop the class conditioning 10% of the time to enable sampling with classifier-free guidance (Ho & Salimans, 2021). We use a guidance scale of $s = 2$ in all our quantitative and qualitative results, unless indicated otherwise. The guidance scale $s$ is defined according to the equation

$$\tilde{\epsilon}^s_\omega(\mathbf{x}_\tau, \mathbf{c}) = \epsilon_\omega(\mathbf{x}_\tau) + s\left(\epsilon_\omega(\mathbf{x}_\tau, \mathbf{c}) - \epsilon_\omega(\mathbf{x}_\tau)\right), \tag{10}$$

using the noise $\epsilon$ prediction formulation (we can easily obtain the noise prediction $\epsilon$ from the $\mathbf{v}$ prediction, which we use for training; see Salimans & Ho (2022)). In the above equation, $\epsilon_\omega(\mathbf{x}_\tau)$ denotes the unconditional score function, $\epsilon_\omega(\mathbf{x}_\tau, \mathbf{c})$ the conditional score function when conditioning on class $\mathbf{c}$, and $\tilde{\epsilon}^s_\omega(\mathbf{x}_\tau, \mathbf{c})$ is the resulting guided score function with guidance scale $s$.

**Compute Limitations and Further Scaling.** Note that due to their noisy training objective, diffusion models have been shown to scale well with more compute and larger batch sizes (Rombach et al., 2021; Karras et al., 2022; Kingma & Gao, 2023). State-of-the-art models on regular, non-3D-aware image synthesis usually use significantly larger batch sizes ($> 1000$) than we do. This suggests that our models in particular on the highly diverse ImageNet dataset could probably improved a lot with more computational resources.

### C.3 MONOCULAR DEPTH

In our pipeline, we leverage geometric cues from a pretrained monocular depth estimation network (Bhat et al., 2023) to supervise the predicted depth $\hat{\mathbf{D}}$ from the autoencoder. Note that the predicted depth is obtained using volume rendering, similarly to Eq. (2) in the main paper

$$\hat{\mathbf{D}}(\mathbf{r}) = \frac{1}{\sum_{j=1}^N w_r^j} \sum_{i=1}^N w_r^i d_r^i \tag{11}$$

where $d_r^i$ denotes the depth of sampling point $i$ along camera ray $r$ and $w_r^i$ is its corresponding rendering weight as defined in Eq. (2) of the main paper. The monocular depth used for supervision, however, is only defined up to scale. Let $\mathbf{D}$ denote the depth predicted by the monocular depth estimator. We first downsample it to match the resolution of the predicted, i.e., rendered depth, which refer to as $\mathbf{D}^{low}$. Next, a scale $s$ and a shift $t$ are computed for each image by solving a least-squares criterion (Eigen et al., 2014; Ranftl et al., 2020)

$$(s, t) = \arg\min_{s,t} \sum_{r \in \mathcal{R}} \left(s\hat{\mathbf{D}}(\mathbf{r}) + t - \mathbf{D}^{low}(\mathbf{r})\right). \tag{12}$$

Defining $\mathbf{h} = (s, t)^T$ and $\mathbf{d}_r = (\hat{\mathbf{D}}(\mathbf{r}), 1)^T$, the closed-form solution is given by

$$\mathbf{h} = \left(\sum_r \mathbf{d}_r \mathbf{d}_r^T\right)^{-1} \left(\sum_r \mathbf{d}_r \mathbf{D}^{low}(\mathbf{r})\right). \tag{13}$$

## D BASELINES

**EG3D\*.** EG3D (Chan et al., 2022) relies on estimated camera poses which are not available for the datasets we consider in this work. Hence, we adapt it to work in view space and remove the need for camera poses. Both the generator and discriminator are originally conditioned on the camera poses. For our version, EG3D\*, we remove the camera pose conditioning and model objects in view space by sampling novel views around the input view $\mathbf{P}_0$ as described in Sec. C.1. For fair comparison, we additionally train a variant of EG3D\* that leverages monocular depth. Specifically, we equip EG3D\*

with the depth discriminator from our pipeline $D_\chi^{depth}$ which compares the rescaled rendered depth with the predictions from a pretrained monocular depth estimatior (Bhat et al., 2023), see Sec. C.3 for more details on the rescaling of the depth.

We follow the training procedure from EG3D (Chan et al., 2022) and ensure that the models train stably by tuning the regularization strength $\lambda$ in the adversarial objective. We use $\lambda = 1$ and $\lambda = 2.5$ for both variants on SDIP Elephants and Horses, respectively. For SDIP Dogs, we found it best to use $\lambda = 2.5$ for EG3D* and $\lambda = 1.0$ EG3D*+ $D_{\text{depth}}$. For the depth discriminator, we set $\lambda_d = 10\lambda$ for all experiments. The models are trained until the discriminator has seen $10M$ images as we find that FID improvements are marginal afterwards. For evaluation, we select the best models in terms of FID.

Note that (Skorokhodov et al., 2023) includes a detailed study on training EG3D without camera poses. As 3DGP clearly outperforms EG3D in this setting, we did not train the original EG3D in canonical space but instead directly compare to 3DGP.

For inversion, we use code kindly provided by the authors of EG3D (Chan et al., 2022). The inversion is performed using PTI (Roich et al., 2023) and consists of two optimization stages. In the first stage, the style code $w$ is optimized to best match the input image. In the second stage, the network parameters are finetuned to further improve reconstruction quality. We observed that the inversion occasionally diverges. For the divergent cases, we reduce the learning rate in the optimization from $10^{-3}$ to $10^{-6}$ finding that this stabilizes the optimization.

**POF3D, 3DGP.** For POF3D (Shi et al., 2023), we use the unpublished code that was kindly provided by the authors to train their model. We follow their training procedure and hyperparameter choices. For 3DGP (Skorokhodov et al., 2023), we trained the models using the publicly available code https://github.com/snap-research/3dgp, which was released shortly before the submission deadline. We found that the training diverges on SDIP Elephants but, as suggested by the authors, were able to restart the model from the last stable checkpoint which then converged. For SDIP Horses training diverges after around $2.5M$ images, even when restarting from the last stable checkpoint, so we report results on the last stable checkpoint. Both POF3D and 3DGP are trained until the discriminator has seen $10M$ images as we observed no or only marginal improvements on FID with longer training.

**StyleNeRF.** We train StyleNeRF (Gu et al., 2022) using the official implementation of the authors https://github.com/facebookresearch/StyleNeRF. On SDIP datasets, we train until the discriminator has seen 20M images, on Imagenet we stop training after 35M images. In both cases, we only observed marginal changes in FID with longer training.

**SceneScape*.** As an additional baseline, we analyzed a combination of a 2D generative model and a 2D inpainting model. We base our implementation on the publicly available code of SceneScape (Fridman et al., 2023) (https://github.com/RafailFridman/SceneScape.git). Specifically, we generate images using the ImageNet checkpoint from LDM (https://github.com/CompVis/latent-diffusion.git). Next, we predict the corresponding depth using ZoeDepth, i.e. using the same pretrained depth estimator as in our approach, and warp the image to a novel view. Lastly, an inpainting model fills the wholes that result from warping. We use an inpainting variant of Stable Diffusion (https://huggingface.co/docs/diffusers/using-diffusers/inpaint#stable-diffusion-inpainting) and provide the warped image, its mask, and the text prompt "a photo of a <class name>" as input.

# E ADDITIONAL RESULTS

**Autoencoder for Compression and Novel-View Synthesis**

Fig. 9 shows further examples of WildFusion and baselines for novel view synthesis on images unseen during training (using GAN-inversion to embed the given images in latent space for EG3D* and EG3D* + $D^{depth}$). We see that our model generally correctly reconstructs the input object and is able to synthesize a high-quality novel view. Moreover, although there are small artifacts, WildFusion also produces plausible geometry. In comparison, the baselines cannot accurately reconstruct the correct object and the geometry is often flat or incorrect.

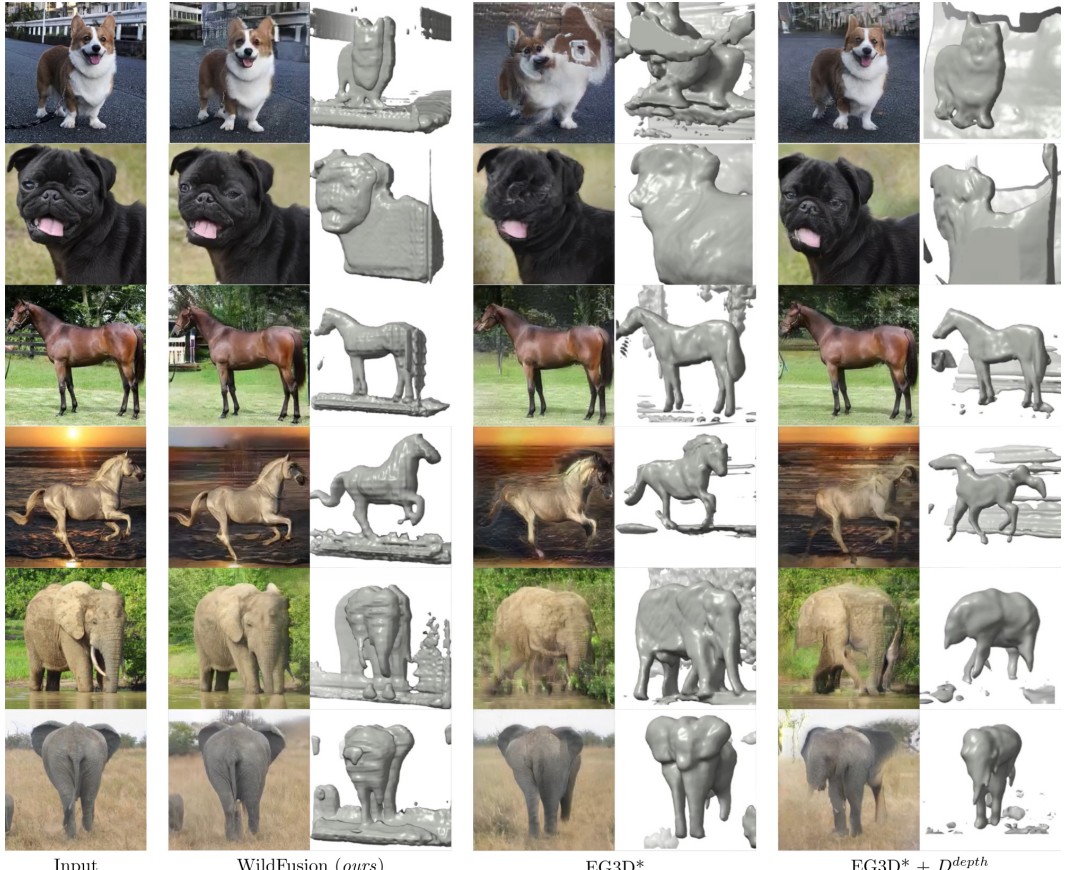

| Input | WildFusion (*ours*) | EG3D* | EG3D* + $D^{depth}$ |

Figure 9: Comparison with baselines for novel view synthesis on images unseen during training. Shown are the input image, a novel view and the geometry extracted with marching cubes. The viewpoints across methods are the same. See the included video for more results.

Fig. 13, Fig. 14, Fig. 15 and Fig. 16 show further novel view synthesis results from WildFusion's 3D-aware autoencoder. The results demonstrate our model's ability to correctly generate novel views, given the encoded input view. All viewpoint changes result in high-quality, realistic outputs.

**3D-Aware Image Synthesis with Latent Diffusion Models**

We first compare WildFusion against the additional baseline SceneScape*. For quantitative analysis, we sample novel views similar to our evaluation by sampling yaw and pitch angles from Gaussian distributions with $\sigma = 0.3$ and $0.15$ radians, using the depth at the center of the image to define the rotation center. With this approach, we get an FID of $12.3$ on ImageNet, compared to $35.4$ for WildFusion. However, as discussed in the main paper, FID only measures image quality and does not consider all important aspects of 3D-aware image synthesis, e.g. 3D consistency. In fact, we observe that the inpainting often adds texture-like artifacts or more instances of the given ImageNet class. We provide some examples in Fig. 10.

To enforce consistency between novel views, we run the full pipeline of SceneScape to fuse the mesh for multiple generated images. For this setting, we sample 9 camera poses by combining yaw angles of $[-0.3, 0., 0.3]$ and pitch angles of $[-0.15, 0., 0.15]$ and iteratively update the mesh by finetuning the depth estimator. We show the final meshes in Fig. 10 (bottom two rows). For all samples we evaluated, we observe severe intersections in the mesh and generally inferior geometry to our approach. We remark that SceneScape's test time optimization takes multiple minutes per sample and a large-scale quantitative evaluation was out of the scope of this work. Our rotating camera movements around a single object are much more challenging, e.g. due to larger occlusions, than the backward motion shown in SceneScape. We hypothesize that this causes SceneScape to struggle more in our setting.

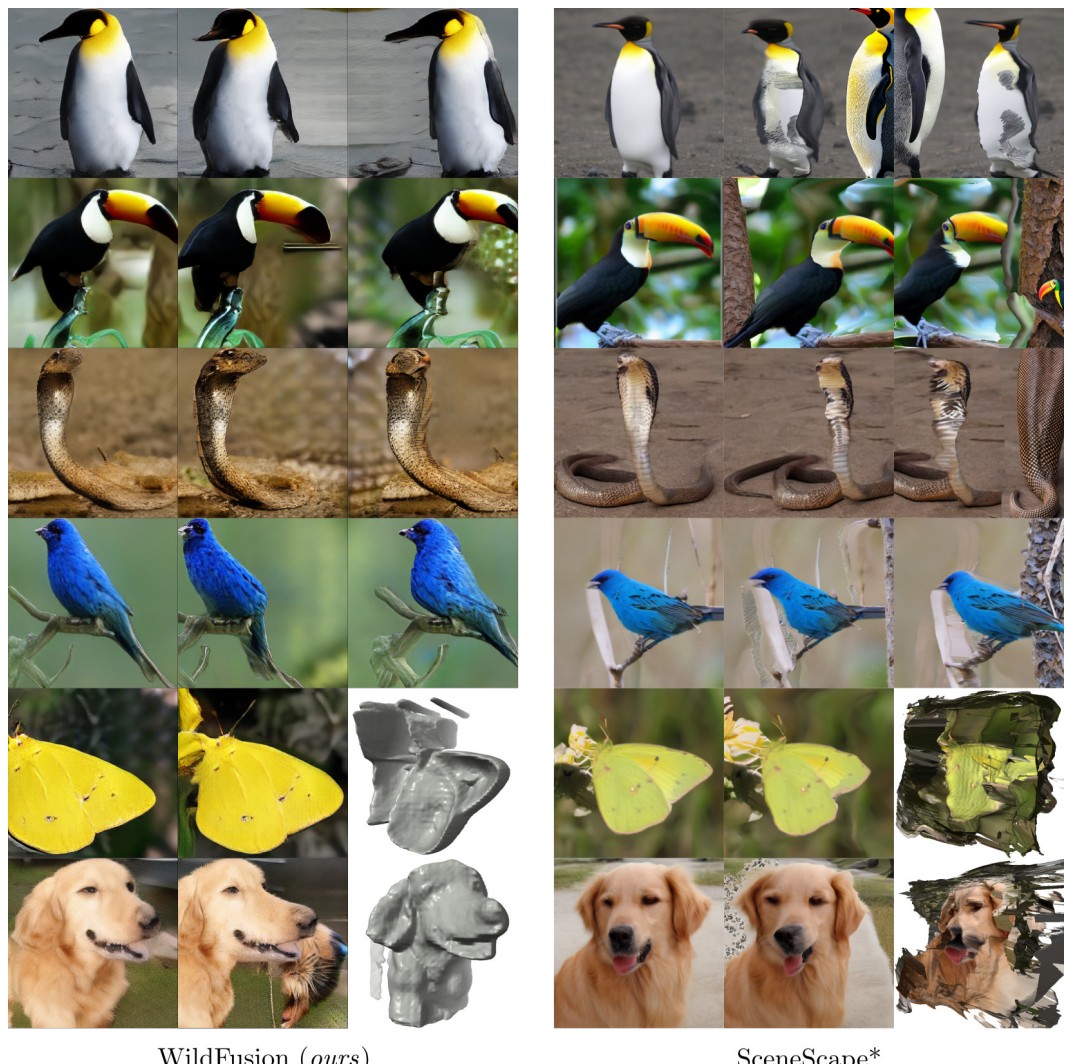

WildFusion (*ours*)          SceneScape*

Figure 10: We compare WildFusion against a variant of SceneScape that combines a 2D generative model with a pre-trained inpainting model. *First for rows*: Leftmost image is input view and next two images are novel views at ±17 degree yaw angles for the two methods. We observe severe inpainting inconsistencies for the SceneScape baseline. *Last two rows*: Leftmost image is again input image, next image is a novel view at −17 degree yaw angle, and the last image shows the extracted geometry/mesh for the two methods). We find that due to the inconsistencies of the inpainting model across views, the fused meshes for SceneScape have severe intersections and overall inferior geometry to WildFusion.

We also include more samples from WildFusion and 3DGP on ImageNet in Fig. 11. While samples from 3DGP look very similar within a class, WildFusion generates diverse samples. Fig. 17, Fig. 18, Fig. 19 and Fig. 20 show further WildFusion results for 3D-aware image synthesis leveraging our latent diffusion model that synthesizes 3D-aware latent space encodings. We can see that all novel samples are high quality and the camera angle changes result in realistic view point changes of the scenes. Note that our model only trains on unposed, single view images and does not need to learn a complex pose distribution because it models objects in view space.

More generated results can be found in the supplementary video, including baseline comparisons and extracted geometries.

**Ablation Studies.**

Fig. 12 visualizes the impact of different configurations on the geometry. For the base config, the model learns a planar geometry which reflects in a low NFS, cf. Tab. 3 in the main paper. Adding a discriminator and the ViT backbone improve geometry significantly but the geometry remains noisy.

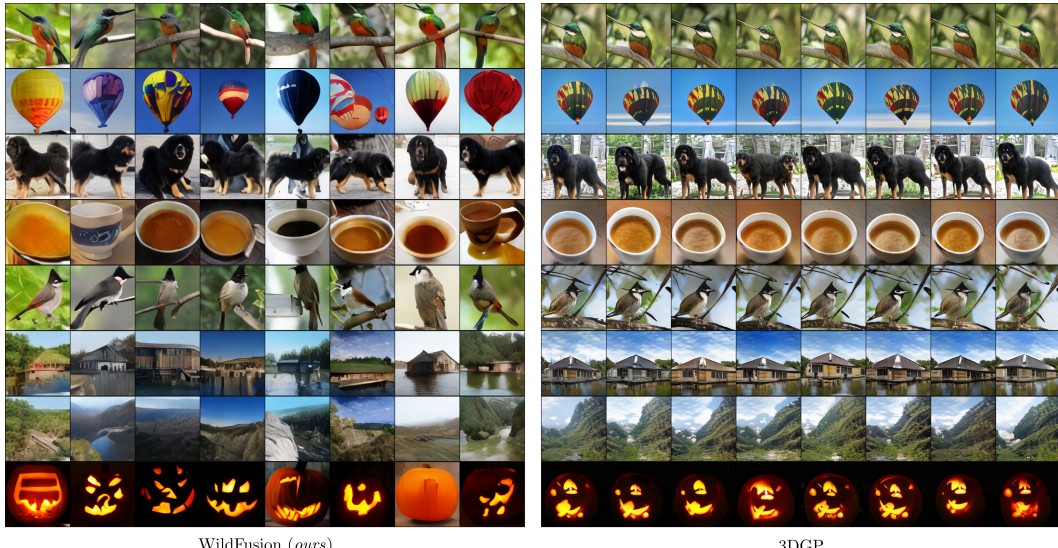

WildFusion (*ours*)                                   3DGP

Figure 11: **Sample Diversity:** Generated samples on ImageNet. Rows indicate class; columns show uncurated random samples. While WildFusion generates diverse samples due to its diffusion model-based framework (*left*), the GAN-based 3DGP (Skorokhodov et al., 2023) has very low intra-class diversity (mode collapse, *right*).

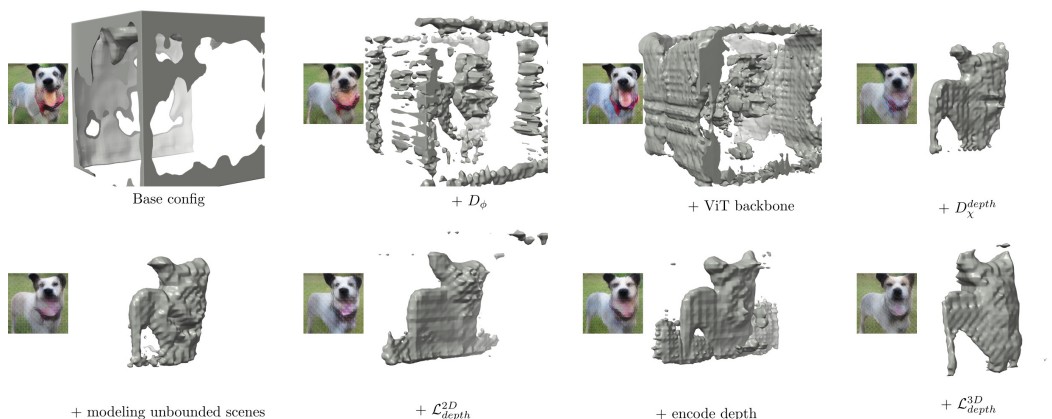

Figure 12: Reconstructions (small) and geometry for different settings in the ablation study. The geometry was extracted by applying marching cubes to the density values of the feature field. We can see an improvement in geometry, as more components are added to the model. Note that the underlying model for this experiment is very small and was used only for the ablation study. It has limited expressivity.

Incorporating geometry cues in the form of monocular depth and a depth discriminator $D_\chi^{depth}$ helps to remove artifacts from the geometry, resulting in a lower NFS. Modeling unbounded scenes with contracted coordinates does not significantly change geometry but improves nvFID, cf. Tab. 3 in the main paper. Further supervision on the rendered depth ($\mathcal{L}_{depth}^{2D}$) does not improve results but as it does not hurt performance either, we kept it in our pipeline. Lastly, both adding depth as an input to the encoder and directly supervising the rendering weights with $\mathcal{L}_{depth}^{3D}$ result in slight improvements in NFS and qualitatively improve geometry. Note how the geometry is less planar when supervising the rendering weights with $\mathcal{L}_{depth}^{3D}$. We remark that the model used in this ablation is much smaller than our main models and has limited expressivity.

| | ↓FID | ↓FID$_{CLIP}$ | ↑NFS | ↑Precision | ↑Recall |
|---|---|---|---|---|---|
| $s = 1$ | 65.1 | 15.3 | 33.6 | 0.58 | 0.16 |
| $s = 1.5$ | 45.2 | 12.9 | 33.7 | **0.60** | 0.19 |
| $s = 2$ | 35.4 | 11.7 | **33.8** | 0.59 | **0.20** |
| $s = 2.5$ | 30.2 | 11.0 | 32.7 | 0.59 | 0.19 |
| $s = 3$ | 28.5 | **10.9** | 32.9 | 0.58 | 0.18 |
| $s = 5$ | **25.5** | 11.6 | 33.6 | 0.53 | 0.13 |
| $s = 10$ | 29.7 | 13.7 | 33.1 | 0.44 | 0.08 |

Table 7: Evaluation on ImageNet with different classifier-free guidance scales $s$.

We further ablate inference with different classifier-free guidance scales (Ho & Salimans, 2021) in Table 7. For better compatibility with previous works, we also evaluated FID on 50K generated images for $s = 3$, which drops from 28.5 on 10K images to 25.3 on 50K images. For computational efficiency, we report FID on 10K images on all other instances throughout the paper.

**Limitations.** Modeling instances in view space alleviates the need for posed images and learned camera distributions. However, it is a very challenging task. This can be seen, for instance, in Fig. 9. It becomes difficult to produce sharp 3D geometry for both baseline models and WildFusion, although WildFusion produces 3D-aware novel-views with high quality and still the most realistic geometry.

Furthermore, as WildFusion is trained with fixed azimuth and polar angle ranges, it is currently not possible to perform novel view synthesis across the full 360°. Increasing the ranges would be an interesting direction for future work. The challenge would lie in producing realistic content when largely unobserved regions become visible after large view point changes. Note, however, that to the best of our knowledge currently there exist no methods that can produce 360° views when trained on the kind of datasets we are training on, which only show a single view per instance, often from a similar front direction.

We observed that the synthesized samples occasionally exhibited plane-like geometry artifacts. Our adversarial loss in the autoencoder, in principle, should avoid this, as it enforces realistic renderings from different viewpoints. We hypothesize that this is due to the autoencoder favoring in rare cases the simple solution of copying the image across triplanes to reduce the reconstruction loss.

Moreover, WildFusion relies on fixed camera intrinsics (see Sec. C.1), which we need to pick ourselves. However, we found that our choice worked well for all three datasets without further tuning. Hence, this is a minor limitation, in particular, compared to methods that work in canonical coordinate systems and need to estimate a pose for each instance (as discussed, this is not possible for many complex, non-aligned datasets). In future work, the camera intrinsics could potentially be learned.

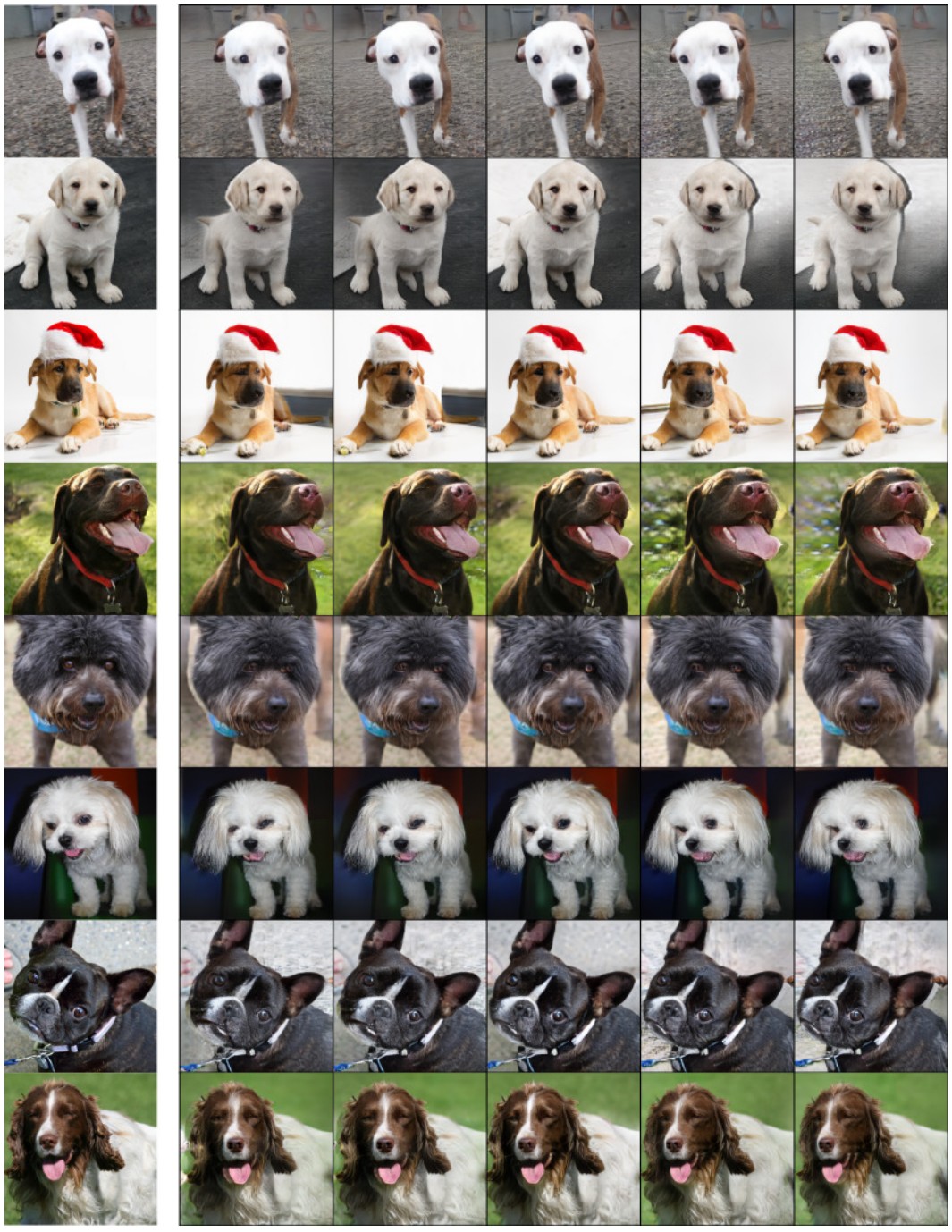

Figure 13: Input images (left column) and novel views from WildFusion's 3D-aware autoencoder for SDIP Dogs. The results span a yaw angle of $40°$.

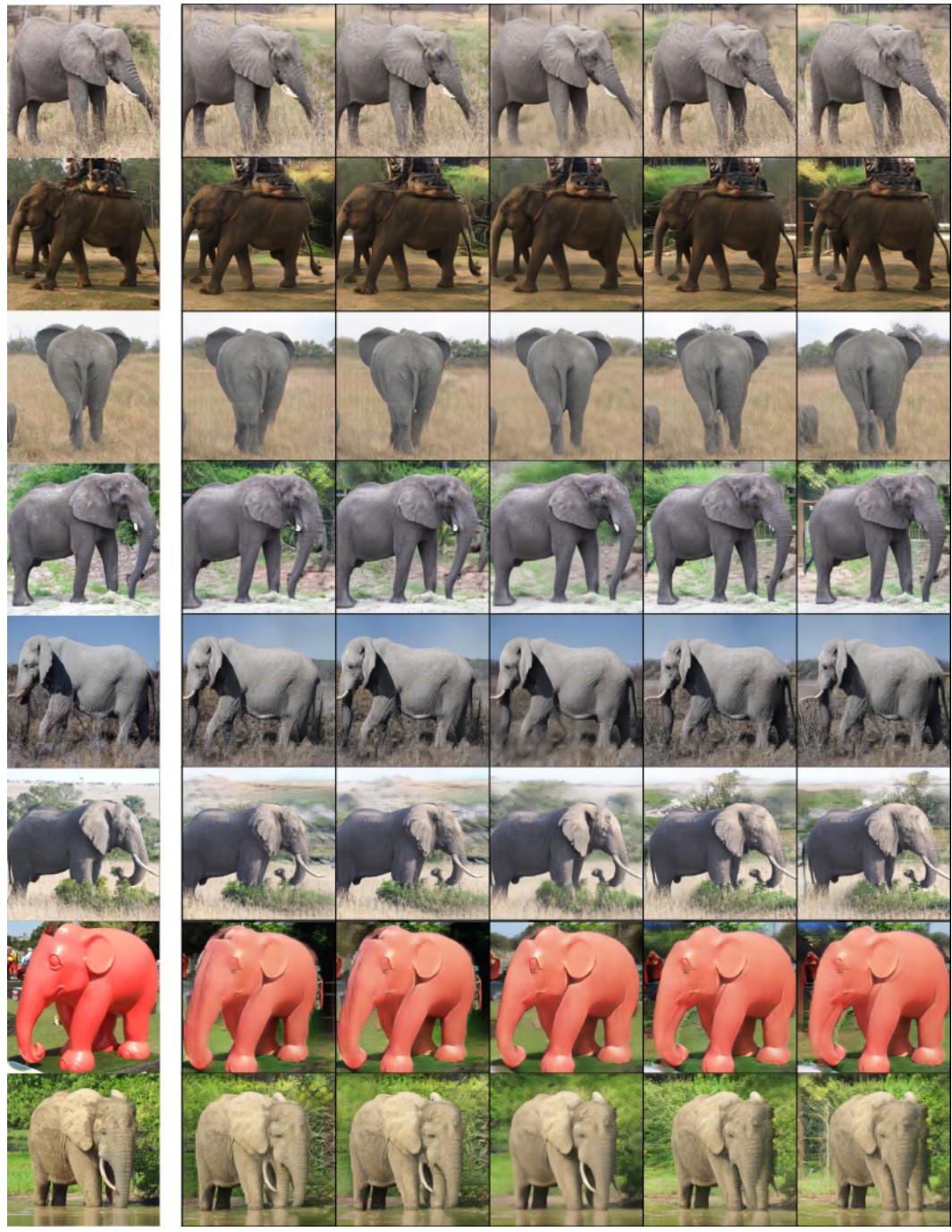

Figure 14: Input images (left column) and novel views from WildFusion's 3D-aware autoencoder for SDIP Elephants. The results span a yaw angle of $40°$.

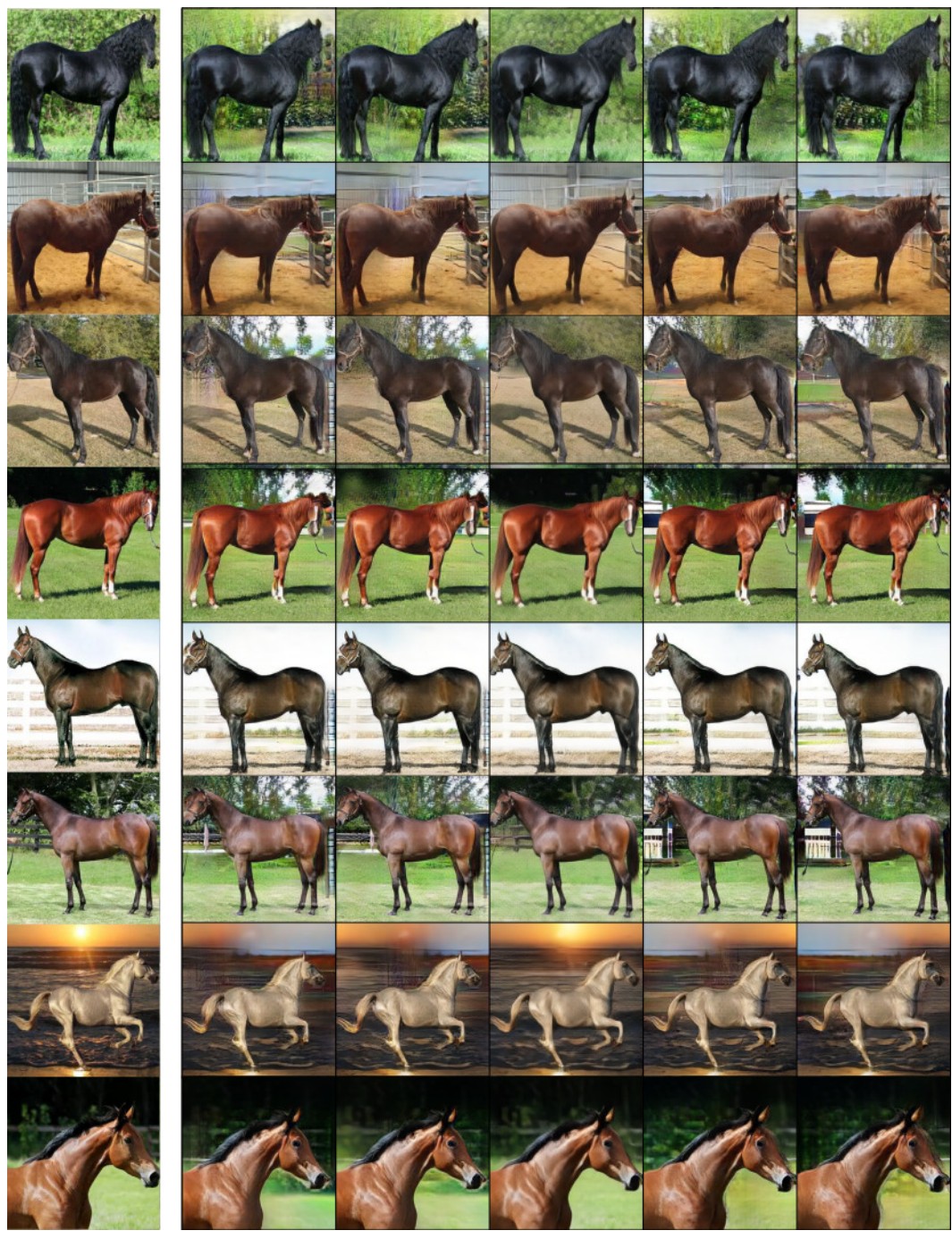

Figure 15: Input images (left column) and novel views from WildFusion's 3D-aware autoencoder for SDIP Horses. The results span a yaw angle of $40°$.

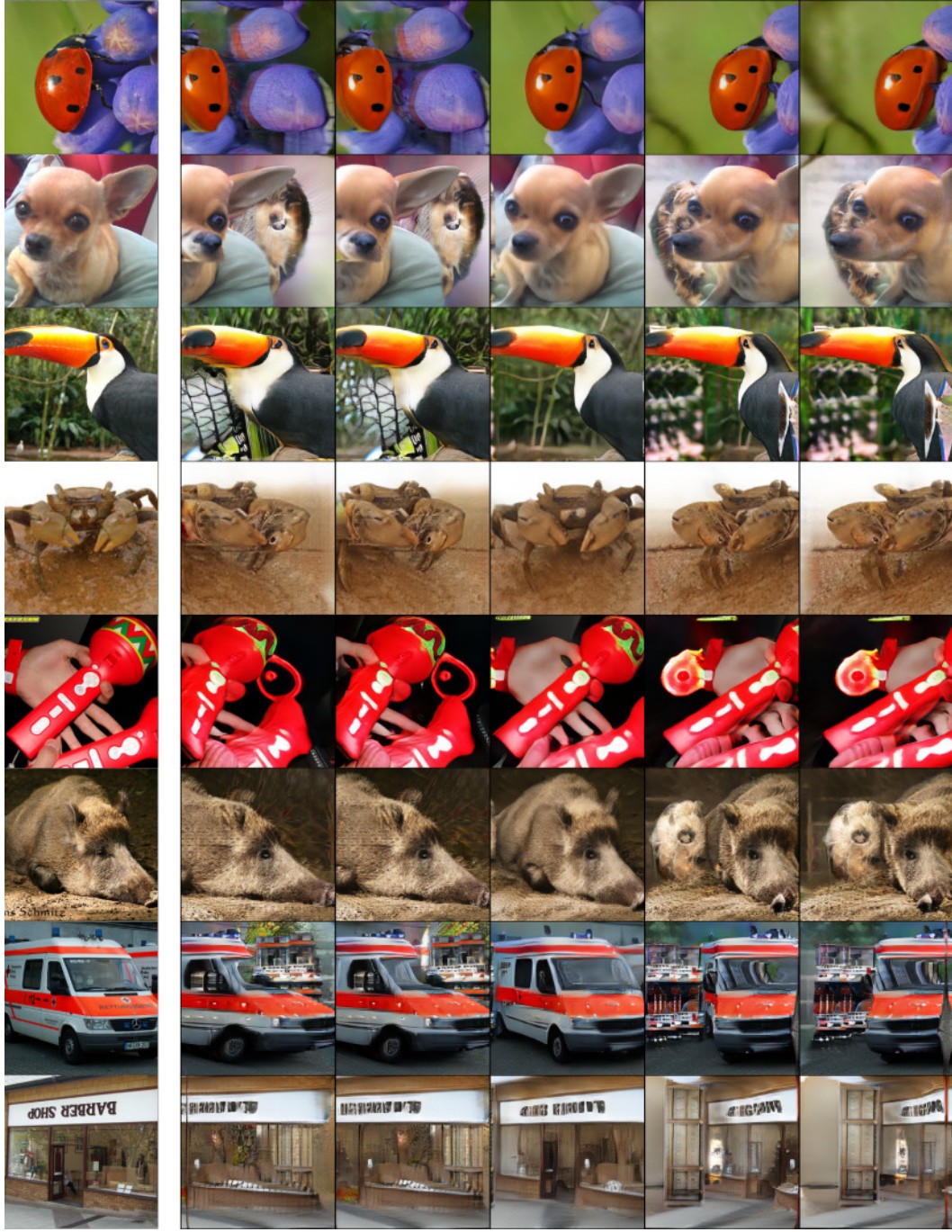

Figure 16: Input images (left column) and novel views from WildFusion's 3D-aware autoencoder for ImageNet. The results span a yaw angle of $40°$.

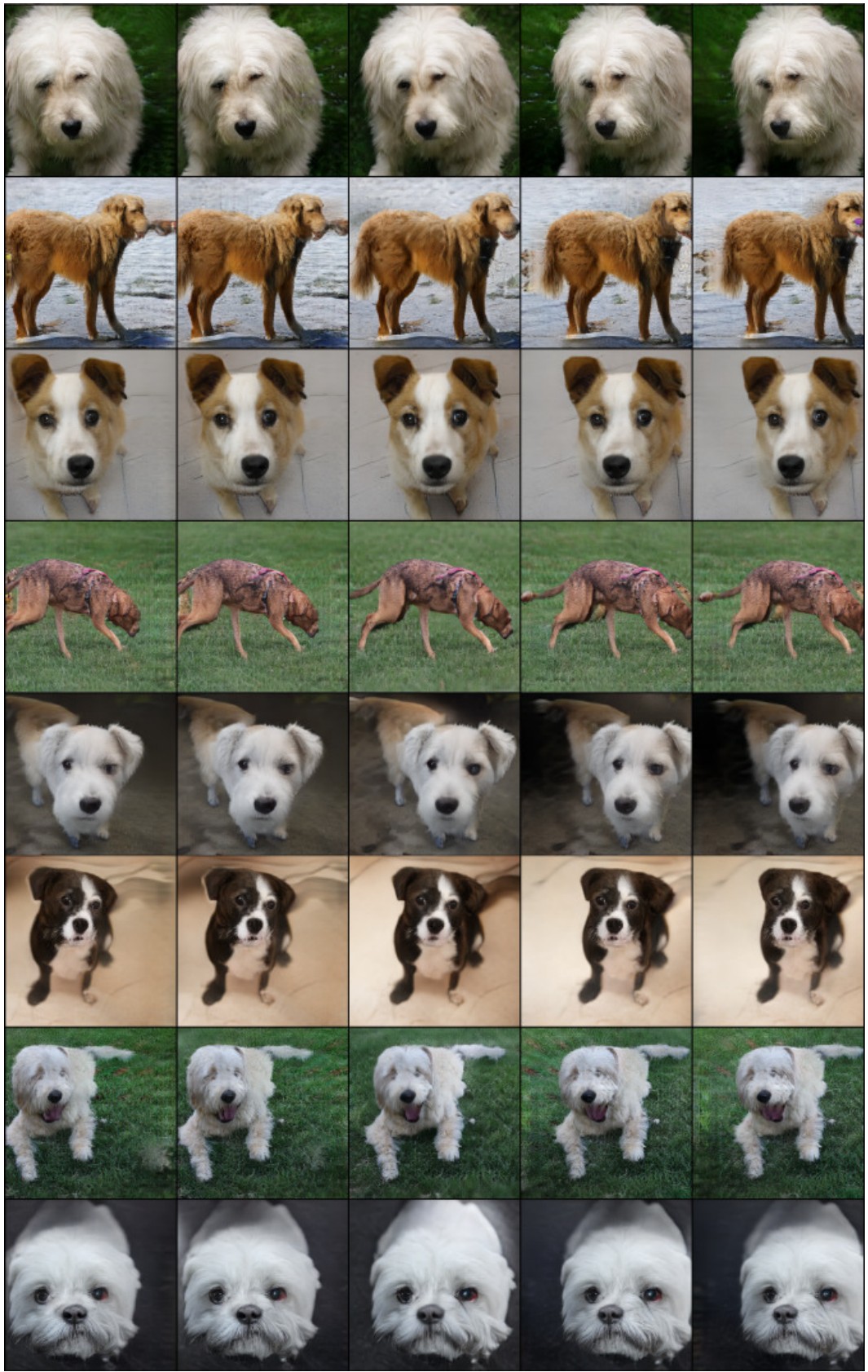

Figure 17: Generated images and novel views from WildFusion for SDIP Dogs. The results span a yaw angle of $40°$.

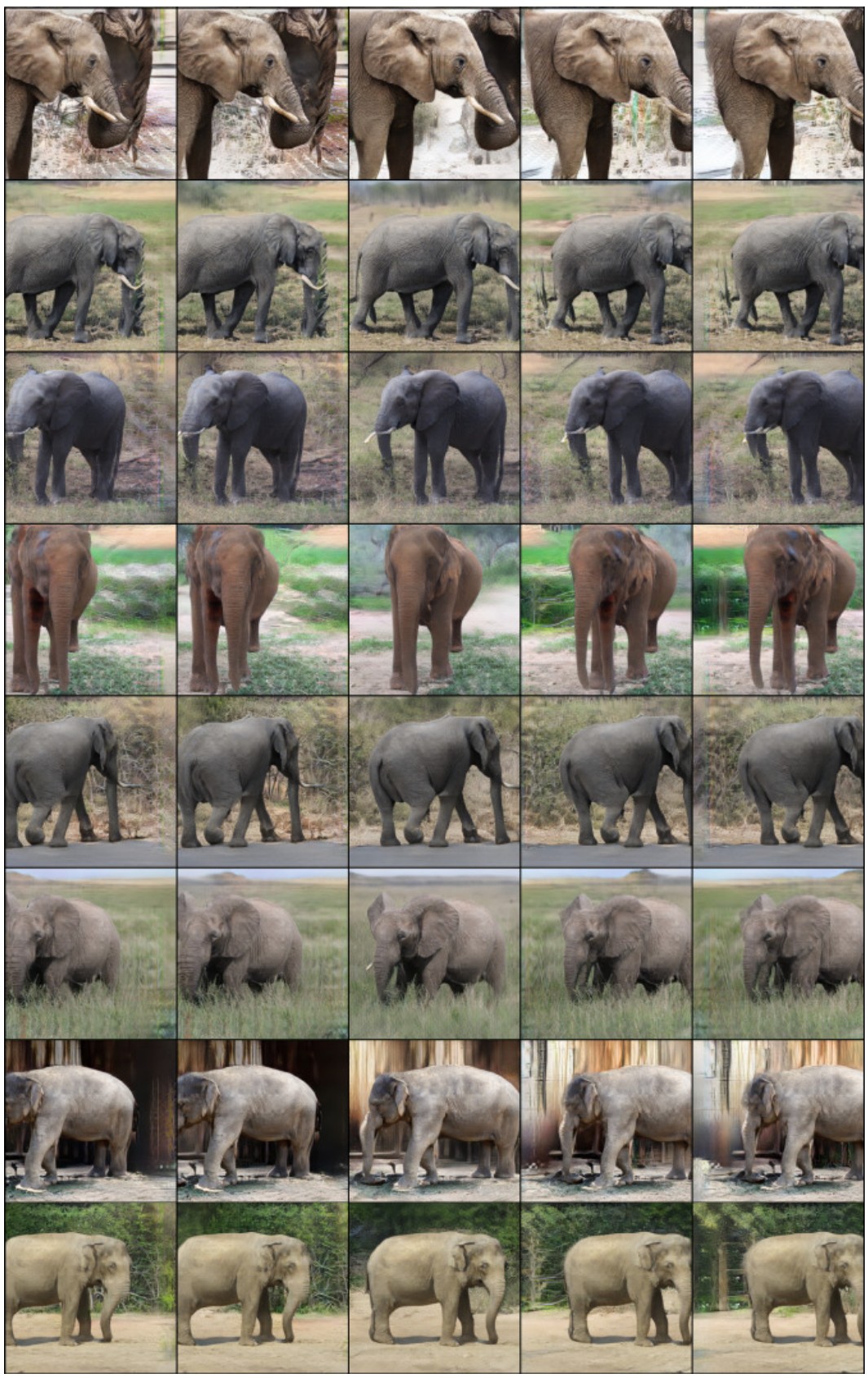

Figure 18: Generated images and novel views from WildFusion for SDIP Elephants. The results span a yaw angle of $40°$.

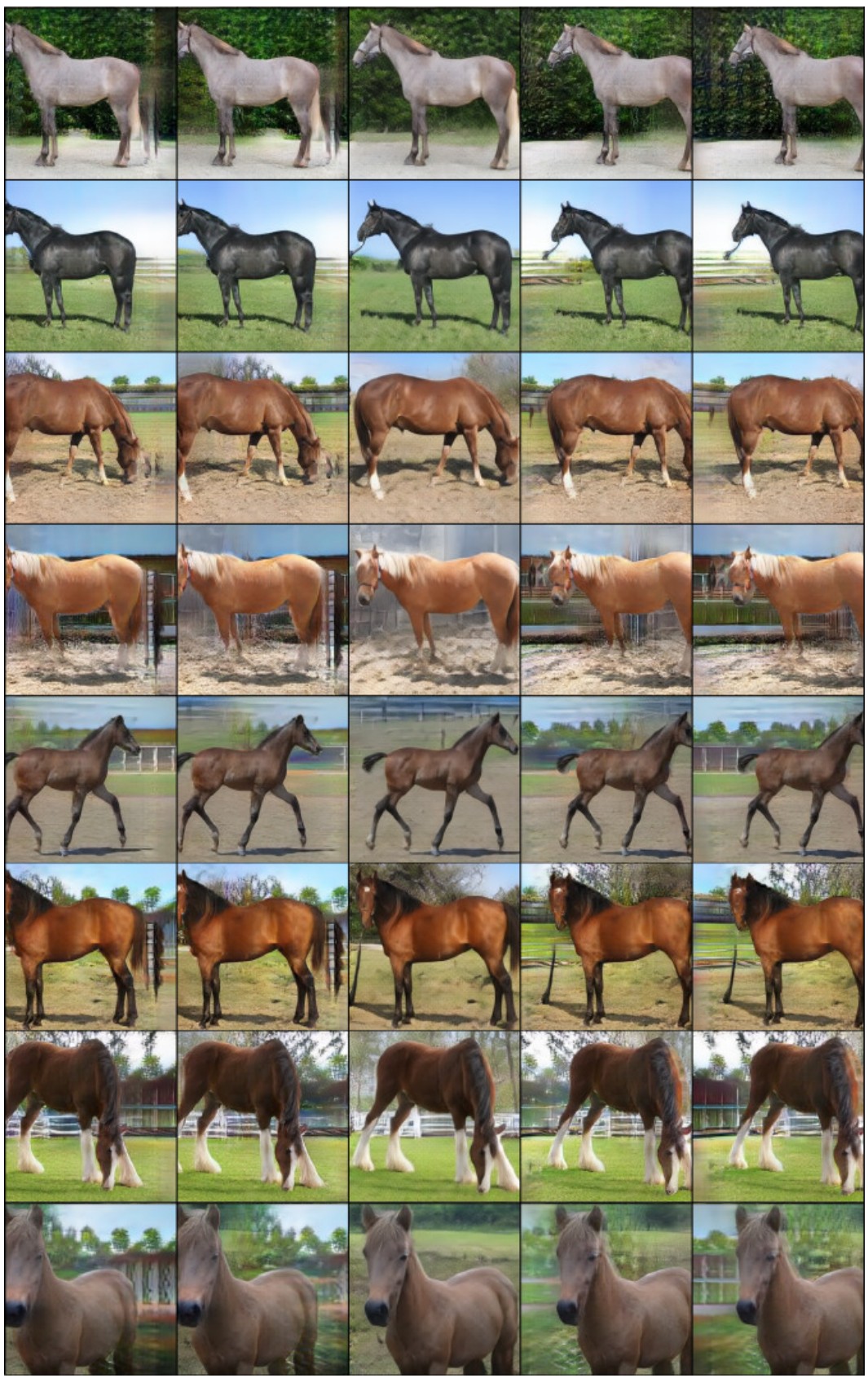

Figure 19: Generated images and novel views from WildFusion for SDIP Horses. The results span a yaw angle of $40°$.

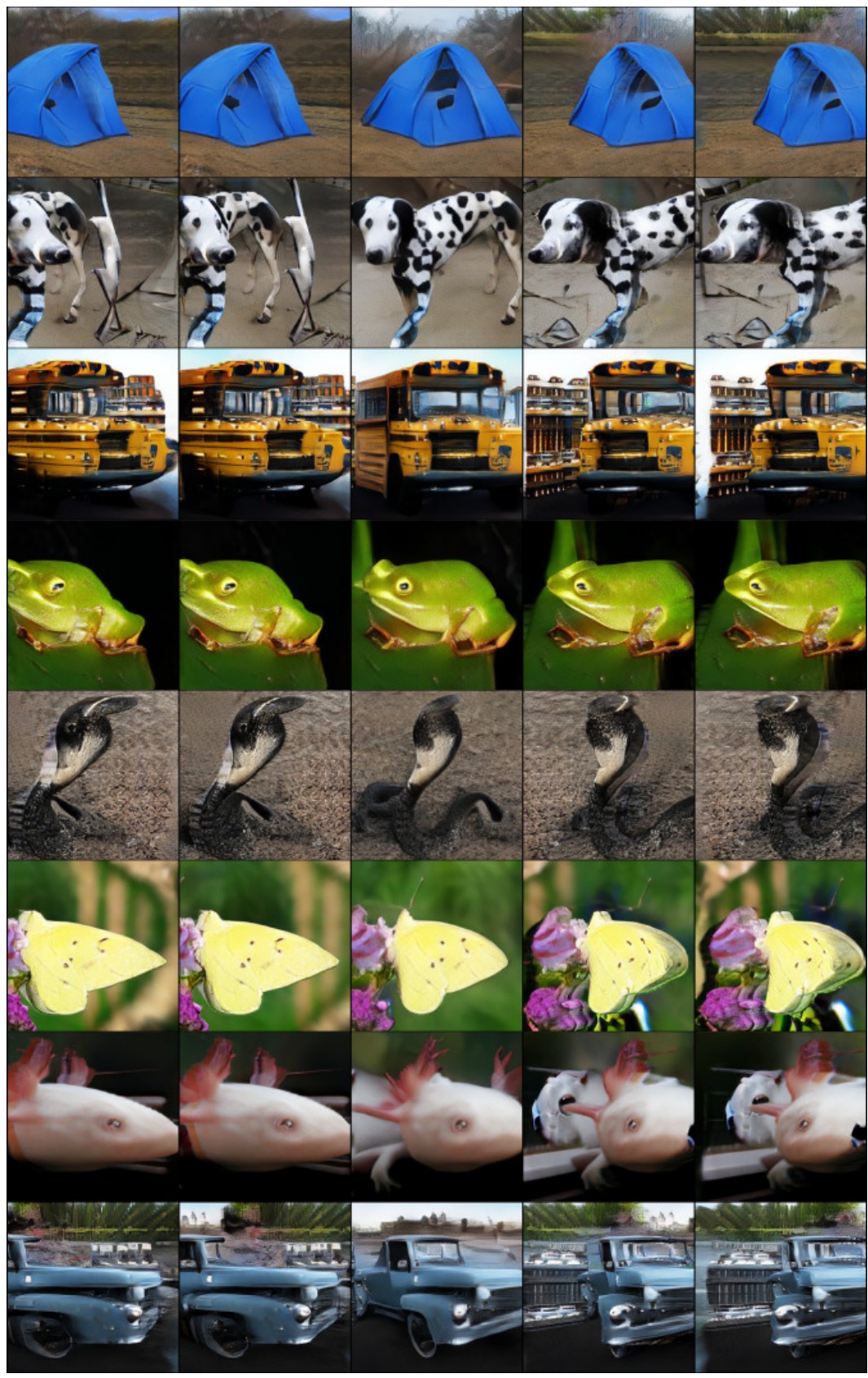

Figure 20: Generated images and novel views from WildFusion for ImageNet. The results span a yaw angle of 40°.

