# OpenReview forum: "WildFusion: Learning 3D-Aware Latent Diffusion Models in View Space"
_ICLR.cc/2024/Conference — ICLR 2024 poster_

### Official Review · Reviewer_CqTg · 2023-10-29

**Soundness:** 3 good
**Presentation:** 3 good
**Contribution:** 3 good
**Rating:** 6
**Confidence:** 4

**Summary:**

This paper presents a method that learns 3D generation from in-the-wild images, e.g., Imagenet. Specifically, this paper proposes to first learn a 3D-aware VAE that compresses images to latent space and decodes latent into 3D representations, i.e., triplane. To facilitate 3D learning from 2D images, especially unposed in-the-wild images, authors proposes to use RGB and depth discriminator for rendered novel views. After learning the 3D-aware latent features, authors uses a standard diffusion model to sample from the latent space. Experiments are performed on Imagenet, as well as three unimodal datasets. Some advantages over GAN-based methods are shown. Ablation study is also performed to show the effectiveness of different model parts.

**Strengths:**

1. A new paradigm is used on the 3D-aware generation on in-the-wild images. The 3D learning happens in the first stage VAE training, where novel view RGB and depth discriminator is used. The second stage latent diffusion model samples from the trained latent space.
2. Advantages of the new paradigm over the GAN-based methods, e.g., 3DGP and EG3D, are shown by quantitative evaluations.

**Weaknesses:**

1. For the first stage of 3D-aware VAE, there are previous methods that propose very similar method. For example, VQ3D and GINA-3D also encodes input images to 3D latents and decode them to the 3D representation of triplane. I would expect authors to give a more detailed explanation of how the proposed method differs from these methods.
2. To continue on the above comment, I would also like to see the performance comparison with VQ3D on both VAE and generator, since the final generation performance largely depends on the performance of the first stage VAE. Therefore, it is important to understand if the performance improvement is from the first or the second stage.
3. I would recommend using 50k, instead of 20k, samples to calculate FID, since it is the standard adopted by most of previous methods, e.g., StyleGAN, VQ3D. It would also allow easier comparison with previous or future work.
4. The visual results shown in Fig. 9 is not very convincing. Flat structure can also be observed from the first two rows.
5. Visual comparison with other methods is not enough. Firstly, I would like to see this comparison in the main paper instead of supplementary material, since it is a very important part for evaluation and can serve as reference when readers try to understand the advantage of the proposed method. Secondly, I would like to see more visual results for mode collapse of GAN-based methods, as mentioned in the third paragraph of section 4.2.

**Questions:**

IVID has shown the ability to generate 360 degree novel view synthesis. I understand that IVID uses a very different approach from this paper. But I would like to hear authors' opinion on learning 360 degree 3D generation with the current pipeline. From my point of view, there is no technical limitation, since you can freely change the novel view camera distribution sampling when training the VAE.

---

> ### Author Response · Authors · 2023-11-21
>
> Thank you for reviewing our work. We appreciate that you consider the soundness, presentation, and contribution of our work “good” (hence, we were surprised by the overall “rejection” rating despite these good scores on all three subcategories). We address your concerns below.
>
> > For the first stage of 3D-aware VAE, there are previous methods that propose very similar method... I would expect authors to give a more detailed explanation of how the proposed method differs from these methods (VQ3D, GINA-3D).
>
> Our initial submission includes a discussion on VQ3D in Appendix A. The main difference to VQ3D is that their autoencoder yields sequence-like latent variables on which an autoregressive transformer is trained while we train a latent diffusion model on feature maps. We extended the discussion of VQ3D in the Appendix of our paper and will move this part of the discussion to the main paper in the camera-ready version. Moreover, it is worth mentioning that VQ3D is a ICCV’23 paper, which took place in early October. According to the ICLR reviewer guidelines (https://iclr.cc/Conferences/2024/ReviewerGuide, Reviewer FAQ, last Q) it is therefore considered concurrent work (anything published at a peer-reviewed venue on or after May 28, 2023 is concurrent).
>
>
> According to our understanding, GINA-3D assumes known camera and LIDAR sensors, which are not given in our setting. Hence, GINA-3D would not be directly applicable in its current form. Also architecturally GINA-3D differs from WildFusion in multiple ways. For instance, it uses a vision transformer encoder and defines its latent space where ta discrete token generative model is trained directly on triplanes, while WildFusion trains a diffusion model on 2D spatial feature maps. GINA-3D only applies losses to the reconstructed image, while our approach uses a discriminator to get guidance on novel views. We included GINA-3D and a discussion in Appendix A in the extended related work section.
>
> > To continue on the above comment, I would also like to see the performance comparison with VQ3D on both VAE and generator
>
> As discussed above, the concurrent VQ3D was only recently published at ICCV’23. Moreover, it does not have any code available for a comparison and a faithful reimplementation is far beyond the scope of this rebuttal. A meaningful comparison to VQ3D is hence impossible and without available code or checkpoints we also cannot evaluate it on all relevant metrics (FID, FID_CLIP, Precision, Recall, NFS).
>
> > I would recommend using 50k.
>
> We evaluated FID50k for WildFusion on ImageNet, which is 25.3. We will include this number in the main paper in the final version.
>
> > Visual comparison with other methods is not enough.
>
> Our initial submission includes 2 figures (Figure 1 and 2)  that illustrate the mode collapse for the strongest GAN-based baseline, 3DGP. We hope that it is understandable that with a limit of 9 pages in total, we cannot add visual comparisons to all baselines in the main paper.
>
> However, we added many more samples of the strongest 3DGP baseline in Appendix D in Figure 14 for more classes. The other GAN-based baselines are known to perform significantly worse than 3DGP (see 3DGP paper, as well as many samples and comparisons on their project page, https://snap-research.github.io/3dgp/ and https://u2wjb9xxz9q.github.io/), hence we did not bother to add even more samples in our paper. However, the quantitative results speak for themselves: Both EG3D and StyleNeRF have very low recall scores (Table 3), indicating similar mode collapse and lack of sample diversity.
>
> > 360 degree novel view synthesis
>
> In theory we could train WildFusion with more drastic camera angle changes, potentially full 360 degree. However, not all ImageNet classes even show objects from all different angles. IVID can likely also only achieve this for some specific classes, and with limited quality and a high inference time compute cost. Consequently, we focussed on the most common setting of most previous work (like the important 3DGP baseline, or EG3D and StyleNeRF), which tackles 3D-aware image generation within a limited camera angle range (IVID is a concurrent work, also published at ICCV’23, with a very different approach, after all. Also note that IVID is discussed in Appendix A). That said, we agree with you that extending WildFusion to full 360 degree 3D synthesis would be an exciting avenue for future work.
>
> We hope that we were able to address your concerns and questions and are happy to discuss should you have remaining questions. If you feel that our rebuttal sufficiently addresses your main concerns, we would like to kindly ask you to consider raising your score accordingly.

---

### Official Review · Reviewer_D5rM · 2023-10-30

**Soundness:** 4 excellent
**Presentation:** 4 excellent
**Contribution:** 2 fair
**Rating:** 8
**Confidence:** 4

**Summary:**

The paper proposes a novel approach to learning a 3D generative model from unposed images in view space.

The core idea of the paper is to first pre-train an auto-encoder that takes an image and its estimated depth map as input, and encodes it into a triplane NeRF. That triplane is then used to re-render the input image and depth as well an one (or several) additional views. The input image and depth are supervised straightforwardly, while the additional view and depth is supervised via an adversarial loss.

Subsequently, the authors propose to train a latent diffusion model on the recovered latent space, enabling unconditional and conditional generative modeling.

**Strengths:**

- Exposure is excellent, the method is exceedingly clear. The overview figure is great.
- The paper is well-motivated and the shortcomings of prior work are clearly highlighted.
- Design choices are clear.
- Baselines are appropriate.
- Ablations are detailed and insightful.

**Weaknesses:**

My core complaint with this paper is that I am not quite sure why you would use this method over a simple depth-warping plus inpainting baseline.

The generated images are of somewhat low quality - they are certainly far behind anything that can be generated with any SOTA 2D generative model.

For any generated image, I could always use the same monocular depth predictor used in this paper to estimate depth, and then warp the image to a novel view. The only challenge would then be holes - which, however, one could easily inpaint, as has been demonstrated in "SceneScape"  (https://arxiv.org/abs/2302.01133).

I would really like to see the following simple baseline:
1. Generate an image with a 2D image generative model.
2. Predict monocular depth with the same model you are currently using.
3. Warp the image to a novel view using the predicted depth.
4. Use an inpainting method such as the one used in "SceneScape".

The only shortcoming here is that this requires warping and in-painting at test time, to render novel views. However, one could easily merge several such in-painted views into a single mesh, which could then be rendered from novel views, similar to what SceneScape does.

Even so, I *do* believe that this paper adds significantly to the literature by clearly formulating the problem, showing how prior methods fail, and producing a method that significantly outperforms prior methods. I think this will spur follow-up work.

**Questions:**

I am overall OK with accepting this paper, as I believe that it is well-written, poses an important problem, and puts forth a reasonable baseline approach.

I would be happy to increase my score if the authors could provide the baseline requested above.


___


I thank the authors for addressing my concerns. I think this additional baseline comparison adds to the paper! I increased my score to 8 and will happily argue for acceptance.

---

> ### Author Response · Authors · 2023-11-21
>
> We would like to thank you for your positive review of our work (“excellent exposure and clarity”, “well-motivated”, “appropriate baselines”, “clear design choices”, “insightful and detailed ablations”). We also appreciate your constructive feedback with respect to the additional experiment you suggested.
>
> We implemented and ran the additional baseline experiment you suggested, building on top of the public implementation of SceneScape (https://github.com/RafailFridman/SceneScape.git).
>
> ------
> **Discussion:**
> We generate images using the ImageNet checkpoint from LDM (https://github.com/CompVis/latent-diffusion.git). Next, we predict the corresponding depth using ZoeDepth, i.e. using the same pretrained depth estimator as in our approach, and warp the image to a novel view. Lastly, an inpainting model fills the holes that result from warping. We use an inpainting variant of Stable Diffusion (https://huggingface.co/docs/diffusers/using-diffusers/inpaint#stable-diffusion-inpainting) and provide the warped image, its mask, and the text prompt “a photo of a <class name>” as input.
>
> For quantitative analysis, we sample novel views similar to our evaluation by sampling yaw and pitch angles from Gaussian distributions with $\sigma = 0.3$ and $0.15$ radians, using the depth at the center of the image to define the rotation center.
> With this approach, we get an FID of 12.3 on ImageNet, compared to 35.4 for WildFusion. However, as discussed in the main paper, FID only measures image quality and does not consider all important aspects of 3D-aware image synthesis, e.g. 3D consistency. In fact, we observe that the inpainting often adds texture-like artifacts or more instances of the given ImageNet class. We provide some examples in Appendix D, Figure 20.
>
> To enforce consistency between novel views, we follow your suggestion and run the full pipeline of SceneScape to fuse the mesh for multiple generated images. For this setting, we sample 9 camera poses by combining yaw angles of $[-0.3, 0., 0.3]$ and pitch angles of $[-0.15, 0., 0.15]$ and iteratively update the mesh by finetuning the depth estimator. We show the final meshes in Appendix D, Figure 20 (bottom two rows). For all samples we evaluated, we observe severe intersections in the mesh and generally inferior geometry to our approach. We remark that SceneScape's test time optimization takes multiple minutes per sample and a large-scale quantitative evaluation was out of the scope of this rebuttal.
>
> Our rotating camera movements around a single object are much more challenging, e.g. due to larger occlusions,  than the backward motion shown in SceneScape. We hypothesize that this causes SceneScape to struggle more in our setting.
>
> -----
>
> We included this discussion in the appendix of our paper, at the very end (Appendix D). For the camera-ready version, we may move this to earlier and add pointers to this experiment in the main paper.
> We hope that we were able to address your concerns. If you have further questions about the implementation of the baseline or any remaining concerns, we are happy to discuss. Otherwise, we would like to kindly ask you to consider raising your score accordingly.

---

### Official Review · Reviewer_HTTZ · 2023-11-01

**Soundness:** 3 good
**Presentation:** 3 good
**Contribution:** 3 good
**Rating:** 8
**Confidence:** 3

**Summary:**

This paper presents a novel method address the challenge of 3D-aware image synthesis for in-the-wild images. A key difference to previous works is that it developed upon a latent diffusion model, which demonstrates more sample diversity to GAN. To enable in-the-wild generation, unlike existing methods that rely on a shared canonical space, the authors propose to model instances in view space. To enable consistent 3D representation, the diffusion model predicts an implicit 3D representation, the triplane representation. The training also leverages monocular depth estimation to further boost 3D accuracy. The experiments show that the proposed work outperforms prior art by a large margin.

**Strengths:**

- The authors leverages latent diffusion model to address the lack of sample diversity in 3D-aware GAN.
- They propose to represent 3D-aware image by an efficient triplane representation.
- The training loss avoids the necessity of multi-view images of the same instance, which makes it easier to train on a much larger amount of data.
- An extensive ablation study to support design choices.

**Weaknesses:**

- 1. This paper only compare with GAN-based methods. It would be more convincing if a comparison to recent diffusion-based methods (GenVS, IVID, VQ3D) is presented.

**Questions:**

See weakness section

---

> ### Author Response · Authors · 2023-11-21
>
> Thank you for reading and reviewing our work and for appreciating our novel approach. While we agree that a comparison with recent diffusion-based methods is interesting, at time of submission no code was available for GenVS, IVID and VQ3D. Also please note that GenVS requires multi-view data for training, which is not available in our setting, so a direct comparison is not possible. Moreover, it is worth mentioning that IVID and VQ3D are ICCV’23 papers, which took place in early October. According to the ICLR reviewer guidelines (https://iclr.cc/Conferences/2024/ReviewerGuide, Reviewer FAQ, last Q) these two papers are therefore considered concurrent work (anything published at a peer-reviewed venue on or after May 28, 2023 is concurrent).

---

### Official Review · Reviewer_US3i · 2023-11-01

**Soundness:** 3 good
**Presentation:** 3 good
**Contribution:** 3 good
**Rating:** 6
**Confidence:** 4

**Summary:**

The paper proposed a two-stage model for learning 3D-aware latent diffusion model in image view space. In the first stage, the authors learn a trasformation: (image, depth) -> latent -> triplane -> NVS images, by applying a GAN loss. Then in the second stage, the authors train a latent diffusion model, which can be directly decoded into a triplane plane NeRF. The learning of the pipeline does not require multi-view data and has shown better results than previous approaches.

**Strengths:**

- The paper is nicely presented. Charts and tables are nicely made and I found the paper easy to read through.

- The two-stage training is interesting. Each step of the pipeline looks reasonable to me.

- The training of the method does not require 3D or multi-view image data.

**Weaknesses:**

- An important work is missing in discussion/comparison. "VQ3D: Learning a 3D-Aware Generative Model on ImageNet", ICCV 2023. The two works are very similar and both works adopt a two-stage learning scheme. The major difference is that VQ3D applies a GAN-based method for both stages.

- I am not sensitive to the quantitative number in the main paper but I saw many NVS results in the supplementary video are distorted. Also, I did not observe a significant visual improvement over the EG3D. I would resort to opinions from other reviewers.

- As the second stage is trained on a latent space obtained by the first stage training, I am concerned that the diffusion generation quality (geometry correctness and image fidelity) is bounded by the GAN-based training. So what is the benefit of introducing the second stage? Easy sampling?

- Strictly speaking, the training involves a large amount of 3D data, which comes from the pre-trained single view depth estimator.

**Questions:**

see above.

---

> ### Author Response · Authors · 2023-11-21
>
> Thank you for reading and reviewing our work. We appreciate that you consider our work well-written and our approach interesting and reasonable. We address your questions in the following.
>
> > An important work (VQ3D) is missing in discussion/comparison.
>
> Our initial submission includes a discussion on VQ3D in Appendix A. The main difference to VQ3D is that their autoencoder yields sequence-like latent variables on which an autoregressive transformer is trained while we train a latent diffusion model on feature maps. Another difference is that VQ3D applies two discriminators on the generated images, one that distinguishes between reconstruction and training image, and another one that discriminates between reconstruction and novel view. We only apply a single discriminator to supervise the novel views and instead have an additional discriminator on the depth. We would like to note that VQ3D was published at ICCV in early October this year and is therefore concurrent work with ours (https://iclr.cc/Conferences/2024/ReviewerGuide, Reviewer FAQ, last Q: anything published at a peer-reviewed venue on or after May 28, 2023 is concurrent) and has no code available for comparison. We extended the discussion of VQ3D in the Appendix of our paper. For the camera-ready version of the paper, we will move this part of the discussion of concurrent work from the Appendix to the main paper.
>
> > I did not observe a significant visual improvement over the EG3D
>
> The strength of diffusion models over GANs becomes most apparent on large, diverse datasets, where GANs often struggle with diversity, coverage of the data distribution and mode collapse. On ImageNet, we clearly outperform EG3D by a very large margin, which is best highlighted in our quantitative metrics (FID: 35.4(ours) vs 111.6(EG3D), recall: 0.19(ours) vs 0.01(EG3D) – this low recall score of 0.01 indicates mode collapse).
>
> >So what is the benefit of introducing the second stage? Easy sampling?
>
> The benefit of introducing the second stage is to be able to use a diffusion model as a generative model. Diffusion models have shown much better coverage and training stability on large, diverse datasets than GANs, which is supported by the recall scores in Table 3. Also note that the first stage autoencoder cannot generate diverse, novel samples on its own, as it always relies on input images for encoding. With the diffusion model in latent space, however, we can generate novel samples from scratch.
>
> > Strictly speaking, the training involves a large amount of 3D data, which comes from the pre-trained single view depth estimator.
>
> Yes, we agree. This is why in the abstract we wrote, “without any direct supervision from multiview images or 3D geometry”. We did indeed not use direct 3D supervision, but only leveraged pre-trained depth estimators. We will add the following sentence to the introduction in the camera ready version to further clarify this:
>
> *To prevent generating flat 3D representations, we leverage cues from monocular depth prediction. While monocular depth estimators are typically trained with multi-view data, we leverage an off-the-shelf pretrained model, such that our approach does not require any direct multi-view supervision for training.*
>
> We remark that such pre-trained depth estimation models are today commonly available and do not represent a major limitation of our method.
>
> We hope that we were able to address your questions and concerns. If so, we would like to kindly ask you to consider raising your score accordingly.

---

### Meta-Review · Area_Chair_EsNR · 2023-12-05

**Metareview:**

The paper proposes a method to train a view synthesis model (with limited view change) from a (monocular) image collection, additionally using a pre-trained monocular depth prediction model. The method involves an adversarially pre-trained view synthesis and autoencoder model, as well as a diffusion model in the latent space. The method performs favorably compared to relevant baselines.

The reviewers are all positive about the paper. They point out that the presentation is good, the method makes sense and performs well. Initially the reviewers had doubts about missing baseline comparisons, but in the rebuttal the authors either provided extra comparisons or clarified that the requested works are concurrent and difficult/impossible to reproduce.

Overall, this is a solid paper and I recommend acceptance.

**Justification For Why Not Higher Score:**

While the method is interesting and novel, neither the technical innovation nor the quality of the results are extremely impressive.

**Justification For Why Not Lower Score:**

This is a solid paper, well written, presenting a reasonable model, and with a thorough experimental evaluation showing that the model performs well.

---

### Decision · Program_Chairs · 2024-01-16

Accept (poster)